# `FedBC`: **Federated Learning Beyond Consensus**

## Abstract

Federated learning (FL) algorithms, such as FedAvg/FedProx, commonly rely on the consensus constraint, enforcing local models to be equal to the global model obtained through the averaging of local updates. However, in practical FL settings with heterogeneous agents, we question the necessity of enforcing consensus. We empirically observe that relaxing consensus constraint improves both local and global performance to a certain extent. To mathematically formulate it, we replace the consensus constraint in standard FL objective with the proximity between the local and the global model controlled by a tolerance parameter $\gamma$, and propose a novel Federated Learning Beyond Consensus (`FedBC`) algorithm to solve it. Theoretically, we establish that `FedBC` converges to a first-order stationary point at rates that matches the state of the art, up to an additional error term that depends on a tolerance parameter $\gamma$. Finally, we demonstrate that `FedBC` balances the global and local model test accuracy metrics across a suite of datasets (Synthetic, MNIST, CIFAR-10, Shakespeare), achieving competitive performance with state-of-the-art.

## 1 Introduction

Federated Learning (FL) has gained popularity as a powerful framework to train machine learning models on edge devices without transmitting the local private data to a central server (McMahan et al., 2017). Mathematically, we can write the FL problem as

$$\min_{\mathbf{x} \in \mathcal{X}} F(\mathbf{x}) := \frac{1}{N} \sum_{i=1}^{N} f_i(\mathbf{x}), \tag{1}$$

where $\mathcal{X} \subset \mathbb{R}^d$ is a compact convex set and $F(\mathbf{x})$ is the sum of $N$ possibly non-convex local objectives $f_i(\mathbf{x})$ which could be stochastic as well $f_i(\mathbf{x}) := \mathbb{E}_{\zeta_i}[f(\mathbf{x}, \zeta_i)]$ with data point $\zeta_i \sim \mathbb{P}(\zeta_i)$ and $\mathbf{x}x$ denotes the model we want to learn such as the weights of neural networks. Following standard FL literature (McMahan et al., 2017; Karimireddy et al., 2020)), we consider that all the devices are connected in a star topology to a central server. The FL problem is challenging because of the heterogeneity across devices which might be due to different sources, such as the local training data sets can have different sample sizes and might not even necessarily be drawn from a common distribution, meaning that $\mathbb{P}(\zeta_i)$ is allowed to be heterogeneous for each device $i$. The goal of standard FL is to train a global model $\mathbf{x}^*$ by solving (1), which performs well or at least uniformly across all the clients (McMahan et al., 2017; Li et al., 2020).

In the presence of data heterogeneity across devices, it is highly unlikely that one global model would work well for all devices. This has been highlighted in (Li et al., 2019), where a large spread in terms of performance of the global model was noted across devices. The requirement of uniform performance of the global model across devices is also connected to *performance disparity* in FL (Li et al., 2019). In FL, the global model is generally constructed from an aggregation of *local models* learned at each device. The simplest is the average of local models in FedAVG (McMahan et al., 2017). When devices' local objectives are distinct, solving (1) can potentially lead to global model which is far away from the local model obtained by solving:

$$\min_{\mathbf{x} \in \mathcal{X}} f_i(\mathbf{x}), \tag{2}$$

for device $i$. For instance, consider the problem of learning "language models" for a cellphone keyboard, where the goal is to predict the next word. FL can be used in such a case to learn a common global model, but a global model might fail to capture distinctive writing styles, as well as the cultural nuances of different users. In such a case, a specific local model [cf. (2)] for each device is required; however, due to sub-sampling error, data at device $i$ might not be sufficient to obtain a reasonable model via only local data. Therefore, there are two competing criteria: *global performance* in terms of (1) evaluated at the global model and a *local performance* evaluated at the local model [cf. (2)]. The notion of global and local models naturally arises in FL and exists in FedAVG (McMahan et al., 2017), FedProx (Li et al., 2020), SCAFFOLD (Karimireddy et al., 2020), etc. Predominately, the focus in the existing literature is either on training only the global model or the local model. Hence we pose the following question:

*"How can one automate the balance between global and local model performance simultaneously in FL?"*

We answer this question affirmatively in this work by developing a novel framework of federated learning beyond consensus (`FedBC`). We propose to consider a problem in which the global objective (1) is primal, which owing to node-separability, allows each device to only prioritize its local objective (2). Then, we introduce a constraint to control the deviation of the local model from the global model with a local hyper-parameter $\gamma_i$ for each device $i$.

**Contributions.** We summarize our *main contributions* as follows:

**(1)** We provide a novel connection between the global and local model improvement and consensus tolerance parameter which is missing from the literature. To characterize it mathematically, we propose a framework of federated learning beyond consensus, which allows us to calibrate the performance of global and local models across devices in FL (cf. 7). This formulation itself is novel for the FL settings.

**(2)** We derive the Lagrangian relaxation of this problem and an instantiation of the primal-dual method, which, owing to node-separability of the Lagrangian, admits a federated algorithm we call `FedBC` (cf. Algo. 1).

**(3)** We establish the convergence of the proposed `FedBC` theoretically and show that the rates are at par with the state of the art. We also illustrate the efficacy of `FedBC` via showing the performance of global and local models on a range of datasets (Synthetic, MNIST, CIFAR-10, Shakespeare).

**Related Works.** Current approaches in literature tend to focus either only on the performance of the global model (McMahan et al., 2017; Li et al., 2020; Karimireddy et al., 2020), or the local model (Fallah et al., 2020; Hanzely et al., 2020), but do not quantitatively calibrate the trade-off between them. Prioritizing global model performance only amongst the individual devices admits a reformulation as a consensus optimization problem (Nedic & Ozdaglar, 2009; Nedic et al., 2010), which gives rise to FedAvg (McMahan et al., 2017). In this context, it is well-known that averaging steps approximately enforce consensus (Shi et al., 2015), whereas one can enforce the constraint exactly by employing Lagrangian relaxations, namely, ADMM (Boyd et al., 2011), saddle point methods (Nedić & Ozdaglar, 2009), and dual decomposition (Terelius et al., 2011). This fact has given rise to efforts to improve the constraint violation of FL algorithms, as in FedPD (Zhang et al., 2021) and FedADMM (Wang et al., 2022). Other approaches involve using model-agnostic meta-learning (Fallah et al., 2020), in which one executes one gradient step as an approximation for (2) as input for solving (1) with objective $\frac{1}{N}\sum_{i=1}^{N} f_i(\mathbf{x} - \alpha\nabla f_i(\mathbf{x}))$. However, it does not explicitly allow one to trade off local and global performance. Several works have sought to balance these competing local and global criteria based upon regularization (Li et al., 2020; Hanzely et al., 2020; T Dinh et al., 2020; Li et al., 2021b). Alternatives prioritize the performance of the global model amidst heterogeneity via control variate corrections (Karimireddy et al., 2020; Acar et al., 2021).

**FL for Global Model Performance:** One of the first popular algorithms to solve the federated learning is FedAvg (McMahan et al., 2017). The idea of FedAvg is to learn a common model for all the devices while updating local models at each device only without communicating local data with the central server. This suffers from the well-known client-drift problem and is further improved in FedProx (Li et al., 2020), SCAFFOLD (Karimireddy et al., 2020), FedDyn and (Acar et al., 2021). Recently, a primal-dual-based approach was proposed called FedPD (Zhang et al., 2021), which also achieves state-of-the-art performance in terms of convergence rate. These works focus on ensuring consensus among the models learned for each

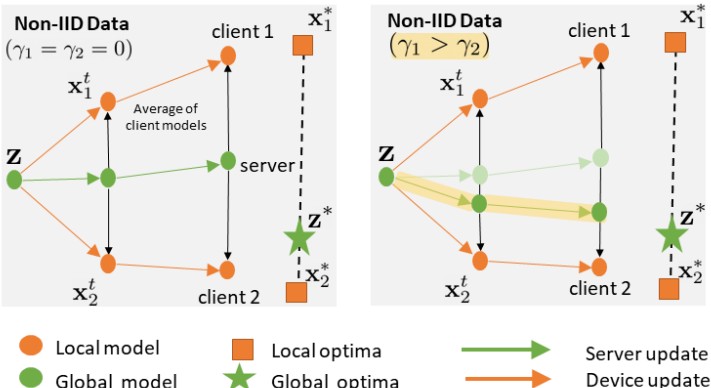

Figure 1: In the left side figure, note that the consensus in standard FL results in averaging at the server, which doesn't allow it to converge to optimal. For the right side figure, the parameter $\gamma_i$ introduces beyond consensus feature and allows the server model to converge to optimal.

device. In this work, we depart from this concept and introduce a novel formulation of federated learning beyond consensus.

**FL for Local Model Performance:** To improve local model performance/personalization, there are mainly two strategies which are employed to introduce personalization into the FL problem, namely: *global model personalization* via meta-learning based methods (Fallah et al., 2020) and *learning personalized models* directly (Tan et al., 2022). In global model personalization, the idea is to learn a common global model in such a way that it performs well when used for the localized objectives after some tuning (Fallah et al., 2020) [some other papers]. Meta-learning is the core idea that is utilized to achieve that. In the second strategy, the focus is on learning specific personalized models for each device to obtain better local performance (T Dinh et al., 2020). The problem with existing personalized methods is that they ignore global performance completely and just focus on local performance. In contrast, in this work, we are trying to find a balance between the two.

## 2 Problem Formulation

In this section, to solve (1) in a federated manner, we consider a consensus reformulation of (1), where each device $i$ is now only responsible for its local copy $\mathbf{x}_i$ of the global model $\mathbf{z}$:

$$\min_{\{(\mathbf{z},\mathbf{x}_i)\in\mathcal{X}\}} \sum_{i=1}^{N} f_i(\mathbf{x}_i) \ \text{ s.t. } \mathbf{x}_i = \mathbf{z}, \ \ \forall i. \tag{3}$$

The linear equality constraints $\mathbf{x}_i = \mathbf{z}$ for all $i$ in (3) enforce consensus among all the devices. To solve (3), one may employ techniques from multi-agent optimization (Nedic & Ozdaglar, 2009; Nedic et al., 2010) and consider localized gradient updates followed by averaging steps, as in FedAvg (McMahan et al., 2017). Setting aside the issue of how sharply one enforces the constraints for the moment, observe that in (3), each device must balance between the two competing global and local objectives. These quantities only coincide when the set of minimizers of the sum is contained inside the set of minimizers of each cost function in the sum. This holds only when the sampling distributions $\mathbb{P}(\zeta_i)$ coincide which is not true for FL in general. Efforts to deal with the gap between the global (1) and local (2) objectives have relied upon augmentations of the local objective, e.g.,

$$f_i(\mathbf{x}_i) + (\mu/2)\|\mathbf{x}_i - \mathbf{z}\|^2 \qquad\qquad\qquad \text{in FedProx,} \tag{4}$$

$$\arg\min_{\boldsymbol{\theta}} f_i(\boldsymbol{\theta}) + (\mu/2)\|\boldsymbol{\theta} - \mathbf{z}\|^2 \qquad\qquad\qquad \text{in pFedMe,} \tag{5}$$

$$f_i(\mathbf{x} - \nabla f_i(\mathbf{x})) \qquad\qquad\qquad \text{in Per-FedAvg.} \tag{6}$$

In the above objectives, observe that a penalty coefficient is introduced to obtain a suitable tradeoff between global and local performance. This relationship is even more opaque in meta-learning, as the tradeoff then

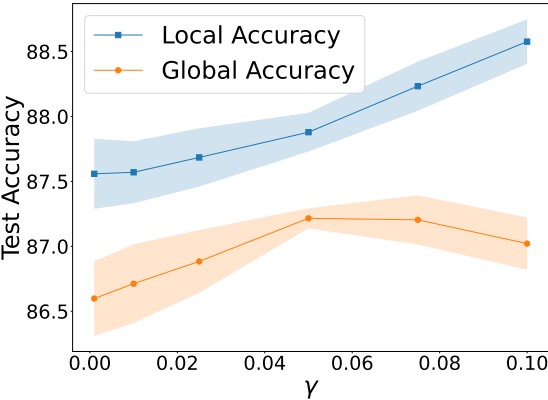

Figure 2: In this figure, $\gamma > 0$ establishes that there is a region $0 < \gamma < 0.05$ to further improve the performance of the global model, as compared to existing FL approaches such as FedAvg, FedProx, SCAFFOLD, etc (where $\gamma = 0$).

depends upon mixed first-order partial derivatives of the local objective with respect to the global model – see (Fallah et al., 2020). Therefore, it makes sense to discern whether it is possible to obtain a methodology to solve for the suitable trade-off between local and global performance while solving for the model parameters themselves. To do so, we reinterpret the penalization in (4) as a constraint, which gives rise to the following problem:

$$\min_{\{(\mathbf{z}, \mathbf{x}_i) \in \mathcal{X}\}} \sum_{i=1}^{N} f_i(\mathbf{x}_i) \ \ \text{s.t.} \ \|\mathbf{x}_i - \mathbf{z}\|^2 \leq \gamma_i, \ \ \forall i \ , \tag{7}$$

for some $\gamma_i \geq 0$. We call this formulation FL beyond consensus because $\gamma_i > 0$ would allow local models to be different from each other and no longer enforces consensus as in (3).

**Interpretation of $\gamma_i$:** The introduction of $\gamma_i$ provides another degree of freedom to the selection of local $\mathbf{x}_i$ and global model $\mathbf{z}$. Instead of forcing $\mathbf{x}_i = \mathbf{z}$ for all $i$ in (3), they both can differ from each other while still solving the FL problem. For instance, consider the example in Fig. 1 (left), where we generalize the example from (Tan et al., 2022) and show (Fig. 1 (right)) that a strictly positive $\gamma_i$ can result in a better global model. Further, as a teaser in Fig. 2, we also note experimentally that $\gamma_i$ calibrates the trade-off between the performance of the local and global model. For simplicity in Fig. 2, we kept $\gamma$ the same for all $i$, and we note that local test accuracy and global test accuracy both increase as we start increasing $\gamma$ from zero, and then eventually global performance starts deteriorating after $\gamma > 0.05$ and local performance is still improving. This makes sense because by making $\gamma$ larger, we are just focusing on minimizing the individual loss functions for each device $i$ than focusing on minimizing the sum. But remarkably, the region between $0 \leq \gamma \leq 0.05$ is interesting because both local and global performance increases, which tells us that $\gamma = 0.05$ is superior to choosing than $\gamma = 0$ as used in the standard FL (McMahan et al., 2017; Li et al., 2020). Hence, this basic experiment in Fig. 2 establishes that there is some room to improve the existing FL models (even if we just focus on the performance of the global model) with a non-zero $\gamma_i$, which has not yet been utilized anywhere to the best of our knowledge. Therefore, this work is the first attempt to show the benefits of using $\gamma > 0$. We further solidify our claims in Sec. 5. Next, we derive an algorithmic tool to solve (7).

## 3 `FedBC`: **Federated Learning Beyond Consensus**

To solve (7), one could consider the primal-dual method (Nedić & Ozdaglar, 2009) or ADMM (Boyd et al., 2011). However, as the constraints [cf (7) are nonlinear, ADMM requires a nonlinear optimization in the inner loop. Thus, we consider the primal-dual method, which may be derived by Lagrangian relaxation of (7)

as:

$$\mathcal{L}(\mathbf{z}, \{\mathbf{x}_i, \lambda_i\}_{i=1}^N) = \frac{1}{N} \sum_{i=1}^N \mathcal{L}_i(\mathbf{z}, \mathbf{x}_i, \lambda_i), \tag{8}$$

where $\mathcal{L}_i(\mathbf{z}, \mathbf{x}_i, \lambda_i) := \left[ f_i(\mathbf{x}_i) + \lambda_i \left( \|\mathbf{x}_i - \mathbf{z}\|^2 - \gamma_i \right) \right]$. Then we alternate between primal minimization and dual maximization. To do so, ideally one would minimize the Lagrangian (8) with respect to $\mathbf{x}_i$ while keeping $\mathbf{z}$ and $\{\lambda_i\}_{i=1}^N$ constant, i.e., at give instant $t$, we solve for $\mathbf{x}_i$ as

$$\mathbf{x}_i^{t+1} = \underset{\mathbf{x}}{\text{argmin}}\, \mathcal{L}_i(\mathbf{x}, \mathbf{z}^t, \lambda_i^t). \tag{9}$$

As local objectives may be non-convex, solving (9) is not simpler than solving (7) for given $\mathbf{z}^t$ and $\lambda_i^t$. To deal with this, we consider an oracle that provides an $\epsilon_i$-approximated solution of the form

$$\mathbf{x}_i^{t+1} = \text{Oracle}_i(\mathcal{L}_i(\mathbf{x}_i^t, \mathbf{z}^t, \lambda_i^t), K_i); \; [K_i\text{-local updates}] \tag{10}$$

where the $\epsilon_i$-approximate solution $\mathbf{x}_i^{t+1}$ is a stationary point of the Lagrangian in the sense of $\|\nabla_{\mathbf{x}_i} \mathcal{L}_i(\mathbf{x}_i^{t+1}, \mathbf{z}^t, \lambda_i^t)\|^2 \le \epsilon_i$. In case of a stochastic gradient oracle, this condition instead may be stated as $\mathbb{E}\left[\|\nabla_{\mathbf{x}_i} \mathcal{L}_i(\mathbf{x}_i^{t+1}, \mathbf{z}^t, \lambda_i^t)\|^2\right] \le \epsilon_i$. We note that any iterative optimization algorithm can be used to perform the $K_i$ local updates. The number of local updates $K_i$ depends upon the accuracy parameter $\epsilon_i$. For instance, in the case of non-convex local objective, a gradient descent-based oracle would need $K_i = \mathcal{O}\left(1/\epsilon_i\right)$ and an SGD-based oracle would require $K_i = \mathcal{O}\left(1/\epsilon_i^2\right)$ number of local steps – see (Wright et al., 1999). A gradient descent-based iteration as an instance of (10) is given in Algorithm 2.

Next, we present the Lagrange multiplier updates initially under the hypothesis that all devices communicate, which we will subsequently relax. In particular, after collecting the locally updated variables at the server, $\mathbf{x}_i^{t+1}$, the dual variable is updated via a gradient ascent step given by:

$$\lambda_i^{t+1} = \mathcal{P}_\Lambda \left[ \lambda_i^t + \alpha(\|\mathbf{x}_i^{t+1} - \mathbf{z}^t\|^2 - \gamma_i) \right], \tag{11}$$

where the dual variable $\lambda_i^{t+1}$ is projected ($\mathcal{P}_\Lambda$ denotes projection operation) onto a compact domain given by $\Lambda := [\lambda_{\min}, \lambda_{\max}]$, where the values of $\lambda_{\min}$ and $\lambda_{\max}$ will be derived later from the analysis. Then, we shift to minimization with respect to the global model variable $\mathbf{z}$, which by the strong convexity of the Lagrangian [cf. (8)] in this variable is obtained by equating $\nabla_{\mathbf{z}} \mathcal{L}(\mathbf{z}, \{\mathbf{x}_i^{t+1}, \lambda_i^{t+1}\}_{i=1}^N) = 0$ and given by

$$\mathbf{z}^{t+1} = \frac{1}{\sum_{i=1}^N (\lambda_i^{t+1})} \sum_{i=1}^N (\lambda_i^{t+1}) \mathbf{x}_i^{t+1}, \tag{12}$$

The server update in (12) requires access to all local models $\mathbf{x}_i$ and Lagrange multipliers $\lambda_i$. To perform the update (12), we use device selection as is common in FL, we uniformly sample a set of $|\mathcal{S}_t|$ devices from $N$ total devices. All the steps are summarized in Algorithm 1.

**Connection to Existing Approaches:** FL algorithms alternate between localized updates and server-level information aggregation. The most common is FedAvg (McMahan et al., 2017), which is an instance of FedBC with $\lambda_i^t = 0$ for all $i$. Furthermore, FedProx is an augmentation of FedAvg with an additional proximal term in the device loss function. Observe that `FedBC` algorithm with $\lambda_i^t = \mu$ for all $i$ and $t$ reduces to FedProx (Li et al., 2020) for (1). Furthermore, for $\gamma_i = 0$ and without device sampling, the algorithm would become a version of FedPD (Zhang et al., 2021). For constant $\lambda_i = c$ and with $K_i = 1$ local GD step, our algorithm reduces to L2GD (Hanzely & Richtárik, 2020), which is limited to convex settings. (Li et al., 2021b) similarly mandates constant Lagrange multipliers and $K_i = 1$.

## 4  Convergence Analysis

In this section, we establish performance guarantees of Algorithm 1 in terms of solving the global [cf. (1)] and the local problem (2). We first state the assumptions:

---

**Algorithm 1 Fed**erated Learning **B**eyond **C**onsensus (`FedBC`)

---

1: **Input**: $T$, $K_i$ for each device $i$, $\gamma_i$ step size parameters $\alpha$ and $\beta$.
2: **Initialize**: $\mathbf{x}_i^0$, $\mathbf{z}^0$, and $\lambda_i^0$ for all $i$.
3: **for** $t = 0$ to $T - 1$ **do**
4:     **Select** a subset of $S$ devices uniformly from $N$ devices , we get $\mathcal{S}_t \in \{1, 2, \cdots, N\}$
5:     Send $\mathbf{z}^t$ to each $j \in \mathcal{S}_t$
6:     **Parallel** loop for each device $j \in \mathcal{S}_t$
7:     Primal update: $\mathbf{x}_j^{t+1} = \text{Oracle}_j(\mathcal{L}_j(\mathbf{x}_j^t, \mathbf{z}^t, \lambda_j^t), K_j)$ according to Algorithm 2
8:     Dual update: $\lambda_j^{t+1} = \mathcal{P}_\Lambda \left[ \lambda_i^t + \alpha(\|\mathbf{x}_i^{t+1} - \mathbf{z}^t\|^2 - \gamma_i) \right]$
9:     Each device $j$ sends $\mathbf{x}_j^{t+1}, \lambda_j^{t+1}$ back to server
10:    **Parallel** for each device $k \notin \mathcal{S}_t$, $x_k^{t+1} = x_k^t$ and $\lambda_k^{t+1} = \lambda_k^t$
11:    **Server** updates

$$\mathbf{z}^{t+1} = \frac{1}{\sum_{j \in \mathcal{S}_t} \lambda_j^{t+1}} \sum_{j \in \mathcal{S}_t} (\lambda_j^{t+1} \mathbf{x}_j^{t+1}) \tag{13}$$

12: **end for**
13: **Output**: $\mathbf{z}^T$

---

**Algorithm 2** Oracle$_i$ in Equation (10) [$K_i$-local updates]

---

1: **Input**: $K_i$, $\gamma_i$, $\beta$, $\mathbf{x}_i^t, \mathbf{z}^t, \lambda_i^t$
2: **Initialize**: $\mathbf{w}_i^0 = \mathbf{x}_i^t$
3: **for** $k = 0$ to $K_i - 1$ for each device $i$ **do**
4:     Update the local model via any optimizer
     **GD optimizer**:
     $\mathbf{w}_i^{k+1} = \mathbf{w}_i^k - \beta \left( \nabla_\mathbf{w} f_i(\mathbf{w}_i^k) + (2\lambda_i^t) \left( \mathbf{w}_i^k - \mathbf{z}^t \right) \right)$
     **SGD optimizer**:
     $\mathbf{w}_i^{k+1} = \mathbf{w}_i^k - \beta \left( \mathbf{g}_i^k + (2\lambda_i^t) \left( \mathbf{w}_i^k - \mathbf{z}^t \right) \right)$
5: **end for**
6: **Output**: $\mathbf{w}_i^{K_i}$

---

**Assumption 4.1.** *The domain $\mathcal{X}$ of functions $f_i$ in (2) is compact with diameter $R$, and at least one stationary point of $\nabla_{\mathbf{x}_i} \mathcal{L}_i(\mathbf{x}_i, \mathbf{z}, \lambda_i) = 0$ belongs to $\mathcal{X}$.*

**Assumption 4.2** (Lipschitz gradients). *The gradient of the local objective $\nabla f_i(\mathbf{x})$ of each device is Lipschitz continuous, i.e., $\|\nabla f_i(\mathbf{x}) - \nabla f_i(\mathbf{y})\| \leq L_i \|\mathbf{x} - \mathbf{y}\|, \forall \mathbf{x}, \mathbf{y} \in \mathcal{X}$ .*

**Assumption 4.3** (Bounded Heterogeneity). *For any device pair $(i, j)$, it holds that $\max_{(a,b) \in \Lambda} \|a \nabla f_i(\mathbf{x}) - b \nabla f_j(\mathbf{x})\| \leq \delta$, for all $\mathbf{x} \in \mathcal{X}$.*

Next, we describe the assumption required when we use stochastic gradients instead of the actual gradients. If we denote the stochastic gradient for agent $i$ as $\mathbf{g}_i$, it satisfies the following assumption.

**Assumption 4.4** (Stochastic Gradient Oracle). *If a stochastic gradient oracle is used at device $i$, then $\mathbf{g}_i$ satisfies $\mathbb{E}[\mathbf{g}_i \mid \mathcal{H}_k] = \nabla f(\mathbf{x}_i)$, and $\mathbb{E}[\|\mathbf{g}_i - \nabla f(\mathbf{x}_i)\|^2 \mid \mathcal{H}_k] \leq \sigma^2$, $\forall i$, where $\mathcal{H}_k$ is defined as filtration or $\sigma$-algebra generated by past realizations $\{\zeta_i^u\}_{u<k}$.*

We note that the Assumptions 4.1-4.4 are standard (Nemirovski et al., 2009). Assumption 4.2 makes sure that the local non-convex objective is smooth with parameter $L_i$. Assumption 4.3 is a version of the heterogeneity assumption considered in the literature (Assumption 3 in (T Dinh et al., 2020)). Assumption 4.4 imposes conditions on the stochastic gradient oracle, particularly unbiasedness and finite variance, which are standard. We are now ready to present the main results of this work in the form of Theorem 4.5. For the convergence analysis, we consider the performance metric $\frac{1}{T} \sum_{t=1}^T \|\nabla f(\mathbf{z}^t)\|^2$ which is widely used in the

literature (McMahan et al., 2017; Li et al., 2020; Karimireddy et al., 2020; T Dinh et al., 2020). Under these conditions, we have the following convergence result.

**Theorem 4.5.** *Under Assumption 4.1-4.4, for the iterates of proposed Algorithm 1, we establish that the global performance satisfies:*

$$\frac{1}{T}\sum_{t=0}^{T-1}\mathbb{E}[\|\nabla f(\mathbf{z}^t)\|^2] \leq \mathcal{O}\left(\frac{M^2 B_0}{T}\right) + \mathcal{O}\left(\frac{M^2\lambda_{\max}}{TN}\sum_{i=1}^{N}\gamma_i\right) + \mathcal{O}\left(\frac{(M+2\lambda_{\max})\epsilon S}{C_0^2 N}\sum_{i=1}^{N}\frac{1}{(\beta L_i)^2}\right)$$
$$+ \mathcal{O}\left(\frac{\delta^2 S}{C_0^2 N}\sum_{i=1}^{N}\frac{1}{(\beta L_i)^4}\right) + \mathcal{O}\left(\frac{1}{N}\sum_{i=1}^{N}\frac{M^2\epsilon}{2L_i}\right) + \mathcal{O}\left(\alpha M^2\right), \qquad (14)$$

*where $B_0$ is the initialization dependent constant, $\epsilon = \max_i \epsilon_i$ is the accuracy with each agent solves the local optimization problem in the algorithm, $\delta$ is the heterogeneity parameter (cf. Assumption 4.3), $\alpha > 0$ is the step size, $\gamma_i$'s are the local parameters, $L_i$ is the Lipschitz smoothness parameter of device $i$, $C_0 := \left(1 - \frac{\lambda_{\max}}{4L_{\min}}\right)$, $L_{\min} = \min_i L_i$, $M_i = L_i + 2\lambda_{\max}$, $M := \max_i M_i$, $\lambda_{\max} < \beta^2 L_{\min}$ is the dual variable upper bound, $\lambda_{\min} > \beta L_{\min}$, $\beta > 1$, $N$ is the number of devices, and we sample $S$ devices uniformly from $N$ devices.*

**Theorem 4.6** (informal version). *Under Assumption 4.1-4.4, for the iterates of proposed Algorithm 1, we establish that the global performance satisfies:*

$$\frac{1}{T}\sum_{t=0}^{T-1}\mathbb{E}[\|\nabla f(\mathbf{z}^t)\|^2] = \mathcal{O}\left(\frac{B_0}{T}\right) + \mathcal{O}(\epsilon) + \mathcal{O}(\delta^2) + \mathcal{O}(\alpha) + \mathcal{O}\left(\frac{1}{NT}\sum_{i=1}^{N}\gamma_i\right), \qquad (15)$$

*where $B_0$ is the initialization dependent constant, $\epsilon = \max_i \epsilon_i$ is the accuracy with each agent solves the local optimization problem in the algorithm, $\delta$ is the heterogeneity parameter (cf. Assumption 4.3), $\alpha > 0$ is the step size, and $\gamma_i$'s are the local parameters.*

An expanded version of Theorem 4.5 and proof is provided in Appendix B. The expectation in (15) is with respect to the randomness in the stochastic gradients and device sampling. The first term is the initialization dependent term, and as long as the initialization $B_0$ is bounded, the first term reduces linearly with respect to $T$ and goes to zero in the limit as $T \to \infty$. This term is present in any state-of-the-art FL algorithm (McMahan et al., 2017; Li et al., 2020; Karimireddy et al., 2020; T Dinh et al., 2020). The second term is $\mathcal{O}(\epsilon)$, which depends upon the worst local approximated solution across all the devices. Note that the individual approximation errors $\epsilon_i$ depend on the number of local iterations $K_i$. This term is also present in most of the analyses of FL algorithms for non-convex objectives. The third term is due to the heterogeneity across the devices and is a specific feature of the FL problem. The fourth term is the step size-dependent term. The last term is important here because that appears due to the introduction of $\gamma_i$ in the problem formulation in (7), and it is completely novel to the analysis in this work. This term decays linearly even if $\gamma_i > 0$ for all $i$. The $\gamma_i$'s are directly affecting the global performance because they are allowing device models to move away from each other, hence affecting the global performance. We remark that for the special case of $\gamma_i = 0$ for all $i$, our result in (4.5) is equivalent to FedPD (Zhang et al., 2021), pFedMe (T Dinh et al., 2020) except for the fact that there is no device sampling in FedPD.

The technical points of departure in the analysis of `FedBC` (cf. Algorithm 1) from prior work are associated with the fact that we build out from an ADMM-style analysis. (Zhang et al., 2021) However, due to non-linear constraint (7), one cannot solve the argmin exactly. This introduces an additional $\mathcal{O}(\epsilon)$ error term that we relate to $K_i$ in (10). This issue also demands we constrain the dual variables to a compact set in (11). Moreover, device sampling for the server update (cf. (13)) is introduced here for the first time in a primal-dual framework, which does not appear in (Zhang et al., 2021). Furthermore, our nonlinear proximity constraints [cf. (7)] additionally permits us to relate the performance in terms of the local objective [cf. (2)] to the proximity to the global model defined in (7) as a function of tolerance parameter $\gamma_i$. We formalize their interconnection in the following corollary.

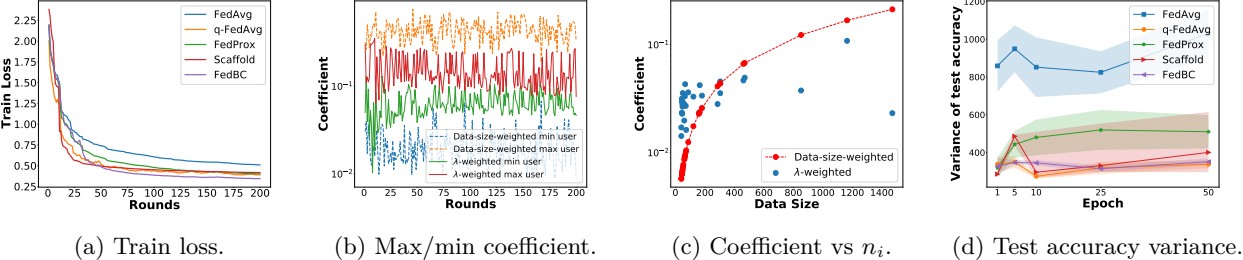

(a) Train loss.     (b) Max/min coefficient.     (c) Coefficient vs $n_i$.     (d) Test accuracy variance.

Figure 3: We use 30 as the total number of users and $E = 5$. (a) We plot global train loss vs the number of rounds and observe that `FedBC` achieves the lowest train loss. (b) We track the user/device with the smallest (min user) or largest (max user) number of data at each round and plot the min or max user's coefficient in computing the global model based on either its local dataset size ($n_i$) or $\lambda_i$. We observe that the magnitude difference between min and max user's coefficient based on $\lambda_i$ is consistently smaller than that based on $n_i$. (c) We plot $\lambda_i$ against $n_i$ for each user at the end of training and observe that users of small dataset sizes (e.g. $< 200$) are able to contribute significantly to the global model in `FedBC`. (d) We show the variance of test accuracy of the global model on each user's local data for different $E$s (shaded area shows standard deviation), and observe that the model of `FedBC` achieves high uniformity.

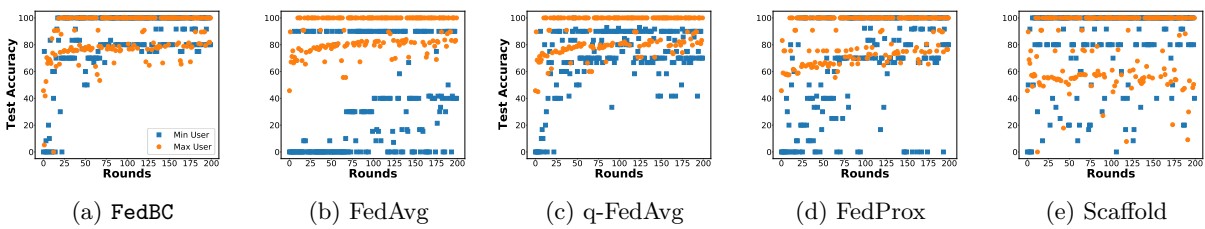

(a) `FedBC`     (b) FedAvg     (c) q-FedAvg     (d) FedProx     (e) Scaffold

Figure 4: Test accuracy of the global model on min and max (defined in Figure 3 caption) users' local data at each round of communication for $E = 5$. The global model of `FedBC` (a) initially has a performance gap on min and max users' data, but the gap is largely eliminated in the end. For FedAvg (b), q-FedAvg (e), FedProx (f), and Scaffold (g), this gap is more apparent and persistent throughout training.

**Corollary 4.7** (Local Performance). *Under Assumption 4.1-4.4, for the iterates of Algorithm 1, we establish that*

$$\frac{1}{T}\sum_{t=1}^{T}\|\nabla f_i(\mathbf{x}_i^t)\|^2 \leq \mathcal{O}\left(\epsilon_i\right) + \alpha^2 \mathcal{O}\left(\left[\sum_{k=0}^{T}\mathbb{I}_{\{\|\mathbf{x}_i^k - \mathbf{z}^{k-1}\|^2 \leq \gamma_i\}}\right]_+\right)^2. \tag{16}$$

An expanded version of Corollary 4.7 and proof is provided in Appendix C. We note that the local stationarity of each client $i$ actually depends on the local $\epsilon_i$ approx error and $\gamma_i$ via a complicated term present in the second term in (16), where $\mathbb{I}$ is an indicator function that is 1 if the condition is not satisfied, and $-1$ otherwise. We note that the term inside the big bracket is larger (worse local performance) for lower $\gamma_i$, and vice versa. Hence we have a relationship between global and local model performance in terms of $\gamma_i$.

## 5 Experiments

In this section, we aim to address the following questions with our experiments: ① *Does the introduction of $\gamma_i$ help `FedBC` to improve global performance compared to other FL algorithms in heterogeneous environments?* ② *Does `FedBC` allow users to have their own localized models and to what extent?* Interestingly, we observed that `FedBC`, with the help of $\gamma_i > 0$, tends to weight the importance of each device equally and hence improves performance disparity as defined by Li et al. (2019). We specifically test the performance disparity of the global model in terms of its performance on user/derive with minimum data (called min user) and device

with maximum data (called max user) in the experiments. Please refer to Appendix D for additional detailed experiments.

Table 1: Synthetic dataset classification global test accuracy for the different numbers of local training epochs, i.e., $K_i = E$ for each device $i$ in Algorithm 1. The $\pm$ shows the standard deviation.

| Algorithm | Epochs | | | | |
|---|---|---|---|---|---|
| | 1 | 5 | 10 | 25 | 50 |
| FedAvg | $83.61 \pm 0.43$ | $83.42 \pm 0.58$ | $83.49 \pm 0.70$ | $83.73 \pm 0.57$ | $82.94 \pm 0.66$ |
| q-FedAvg | $87.12 \pm 0.25$ | $86.76 \pm 0.15$ | $86.46 \pm 0.28$ | $86.76 \pm 0.12$ | $86.71 \pm 0.12$ |
| FedProx | $86.23 \pm 0.42$ | $85.59 \pm 0.37$ | $85.34 \pm 0.61$ | $85.00 \pm 0.63$ | $85.34 \pm 0.48$ |
| Scaffold | $83.84 \pm 0.09$ | $82.95 \pm 0.30$ | $83.48 \pm 0.21$ | $83.60 \pm 0.21$ | $82.80 \pm 0.62$ |
| FedBC | $\mathbf{87.83 \pm 0.35}$ | $\mathbf{87.48 \pm 0.18}$ | $\mathbf{87.43 \pm 0.20}$ | $\mathbf{86.99 \pm 0.11}$ | $\mathbf{87.26 \pm 0.12}$ |

**Experiment Setup.** The synthetic dataset is associated with a 10-class classification task, and is adapted from (Li et al., 2020) with parameters $\alpha$ and $\beta$ controlling model and data variations across users (see Appendix D.1 for details). For real datasets, we use MNIST and CIFAR-10 for image classification. MNIST and CIFAR-10 datasets consist of handwritten digits and color images from 10 different classes respectively (Krizhevsky et al., 2009) (LeCun et al., 1998). We denote $C$ to represent the most common number of classes in users' local data (see Appendix D.3 for details). To evaluate the global performance of FedBC, we compare it with 4 other FL algorithms, i.e., FedAvg (McMahan et al., 2017), q-FedAvg (Li et al., 2019), FedProx (Li et al., 2020), and Scaffold (Karimireddy et al., 2020). We use the term *global accuracy* while reporting the performance of the global model ($\mathbf{z}^t$) on the entire test dataset and use the term *local accuracy* while reporting the performance of each device's local model ($\mathbf{x}_i^t$) using its own test data and take the average across all devices.

**Selection of $\gamma_i$:** Since we do not know the optimal $\gamma_i$ for each user $i$ apriori, for experiments, we initialize them to be 0, and propose a heuristic to let the device decide its own $\gamma_i$. To achieve that, we observe $\gamma_i$ participating in the Lagrangian defined in (8) and which defines a loss with respect to primal variables, and we want to minimize it. Hence, we take the derivative of the Lagrangian in (8) with respect to $\gamma_i$, and perform a gradient-descent update for $\gamma_i$. Interestingly, we note that the derivative of $\gamma_i$ is $-\lambda_i$, which means that $\gamma_i$ tends to always increase when gradient descent is performed. This implies that initially, each device's local model remains closer to the global model (similar to standard FL), but gradually incentivizes moving away from the global model to improve the overall performance. Experimentally, we show that this heuristic works very well in practice. (see Appendix D.1 for additional details).

**Synthetic Dataset Experiments:** We start by presenting the global accuracy results of the synthetic dataset classification task in Table 1. Note that FedBC outperforms all other algorithms for different numbers of local training epochs $E$. Figure 3 provides empirical justifications for such remarkable performance. Figure 3a shows that FedBC achieves a lower training loss than others, whereas algorithms such as FedAvg plateaus at an early stage. Next, to understand the calibrating behavior of FedBC, we compare the contributions from min and max users' local models (defined in Figure 3 caption) in updating the global model $\mathbf{z}^t$ via (13). To this end, we plot their $\lambda_i$s at the end of each communication round in Figure 3b. It is evident that FedBC is significantly less biased towards the min user. The min user's coefficient eventually catches that of the max user for FedBC, and the difference between them is one order of magnitude less than that of the data-size-based coefficient. This enables FedBC to be *better in terms of performance disparity* as compared to other algorithms. Figure 3c shows the distribution of $\lambda$- coefficients at the end of training for devices of different data sizes. Interestingly, the max user's coefficient is almost the same as those of users of small data sizes. In fact, the coefficients of users of data size less than 300 for FedBC are consistently larger than their data-size-based counterparts, and vice versa for data sizes greater than 300. Lastly, Figure 3d shows the model of FedBC achieves a high uniformity in test accuracy over users' local data at different values of $E$.

To further emphasize the performance disparity aspect of FedBC, we plot the test accuracy of the global model on min and max users' local data throughout the entire communication in Figure 4. We observe that the global model performs better on max user's data than on min user's data in general for all algorithms. Most importantly, FedBC demonstrates a superior advantage in reducing this performance gap. After 100 rounds of

Table 2: Global model performance of `FedBC` and other baselines on CIFAR-10 classification ($E = 5$). All colored cells denote the proposed algorithms.

| Classes | Algorithm | Power Law Exponent | | | | |
|---|---|---|---|---|---|---|
| | | 1.1 | 1.2 | 1.3 | 1.4 | 1.5 |
| C = 1 | FedAvg | $50.15 \pm 0.58$ | $50.66 \pm 1.88$ | $57.23 \pm 0.74$ | $53.69 \pm 2.24$ | $60.82 \pm 0.85$ |
| | q-FedAvg | $49.17 \pm 1.38$ | $49.32 \pm 1.64$ | $57.35 \pm 1.05$ | $54.40 \pm 1.57$ | $60.56 \pm 1.19$ |
| | FedProx | $49.92 \pm 1.16$ | $50.21 \pm 2.00$ | $57.14 \pm 0.60$ | $56.30 \pm 1.95$ | $60.57 \pm 0.65$ |
| | Scaffold | $46.86 \pm 2.03$ | $36.55 \pm 2.63$ | $37.81 \pm 2.50$ | $30.99 \pm 1.23$ | $36.08 \pm 3.25$ |
| | Per-FedAvg | $45.93 \pm 0.86$ | $37.03 \pm 6.22$ | $56.43 \pm 2.35$ | $52.82 \pm 1.98$ | $56.31 \pm 5.52$ |
| | pFedMe | $47.18 \pm 1.28$ | $43.69 \pm 1.38$ | $50.76 \pm 1.44$ | $45.12 \pm 1.92$ | $50.19 \pm 2.24$ |
| | FedBC | $\mathbf{50.35 \pm 0.91}$ | $\mathbf{55.25 \pm 1.27}$ | $\mathbf{58.93 \pm 1.52}$ | $\mathbf{58.10 \pm 1.56}$ | $\mathbf{61.12 \pm 1.24}$ |
| C = 2 | FedAvg | $57.10 \pm 0.85$ | $56.94 \pm 1.66$ | $57.67 \pm 2.76$ | $32.87 \pm 2.82$ | $58.59 \pm 3.14$ |
| | q-FedAvg | $\mathbf{57.20 \pm 0.68}$ | $\mathbf{58.29 \pm 1.67}$ | $57.71 \pm 2.09$ | $57.10 \pm 2.44$ | $57.90 \pm 3.27$ |
| | FedProx | $57.18 \pm 1.21$ | $57.64 \pm 1.64$ | $57.98 \pm 1.73$ | $39.99 \pm 4.18$ | $58.63 \pm 2.91$ |
| | Scaffold | $55.51 \pm 1.79$ | $24.48 \pm 3.34$ | $36.69 \pm 2.78$ | $41.79 \pm 0.42$ | $32.18 \pm 9.18$ |
| | Per-FedAvg | $55.29 \pm 0.82$ | $54.67 \pm 1.88$ | $54.64 \pm 1.96$ | $39.66 \pm 6.94$ | $60.16 \pm 1.46$ |
| | pFedMe | $51.45 \pm 0.44$ | $46.80 \pm 2.04$ | $47.98 \pm 1.46$ | $30.81 \pm 3.22$ | $49.25 \pm 1.68$ |
| | FedBC | $56.45 \pm 1.05$ | $55.64 \pm 1.45$ | $\mathbf{58.02 \pm 2.10}$ | $\mathbf{60.92 \pm 2.43}$ | $\mathbf{64.20 \pm 2.13}$ |
| C = 3 | FedAvg | $64.19 \pm 2.06$ | $53.83 \pm 3.38$ | $57.54 \pm 1.17$ | $60.96 \pm 0.95$ | $58.91 \pm 2.76$ |
| | q-FedAvg | $62.76 \pm 1.53$ | $61.37 \pm 2.06$ | $61.80 \pm 1.73$ | $63.40 \pm 2.16$ | $64.01 \pm 1.96$ |
| | FedProx | $\mathbf{64.40 \pm 1.80}$ | $54.81 \pm 2.16$ | $57.28 \pm 1.90$ | $61.66 \pm 0.42$ | $59.85 \pm 2.81$ |
| | Scaffold | $61.46 \pm 1.86$ | $54.04 \pm 6.99$ | $38.38 \pm 2.80$ | $30.75 \pm 5.68$ | $33.04 \pm 9.85$ |
| | Per-FedAvg | $61.83 \pm 1.74$ | $52.24 \pm 1.84$ | $58.16 \pm 0.61$ | $60.36 \pm 1.96$ | $59.27 \pm 2.46$ |
| | pFedMe | $53.52 \pm 1.59$ | $42.92 \pm 1.64$ | $52.50 \pm 1.79$ | $54.39 \pm 1.91$ | $53.26 \pm 1.24$ |
| | FedBC | $63.43 \pm 2.55$ | $\mathbf{62.90 \pm 2.26}$ | $\mathbf{64.87 \pm 1.44}$ | $\mathbf{66.23 \pm 0.65}$ | $\mathbf{66.80 \pm 1.07}$ |

Table 3: Local model performance of `FedBC` and other baselines on CIFAR-10 classification ($E = 5$). All colored cells denote the proposed algorithms (see Algorithm 3 in the appendix for Per-FedBC).

| Classes | Algorithm | Power Law Exponent | | | | |
|---|---|---|---|---|---|---|
| | | 1.1 | 1.2 | 1.3 | 1.4 | 1.5 |
| C = 1 | Per-FedAvg | $\mathbf{99.92 \pm 0.15}$ | $\mathbf{85.58 \pm 0.66}$ | $\mathbf{94.99 \pm 0.37}$ | $91.10 \pm 1.90$ | $85.09 \pm 1.42$ |
| | pFedMe | $86.28 \pm 0.59$ | $72.05 \pm 0.35$ | $81.21 \pm 0.27$ | $82.46 \pm 0.65$ | $76.94 \pm 0.82$ |
| | FedBC | $91.96 \pm 0.70$ | $77.52 \pm 0.60$ | $84.87 \pm 1.01$ | $86.15 \pm 0.43$ | $81.31 \pm 0.22$ |
| | FedBC -FineTune | $97.36 \pm 0.22$ | $84.54 \pm 0.17$ | $91.89 \pm 0.44$ | $87.18 \pm 0.21$ | $82.01 \pm 0.17$ |
| | Per-FedBC | $99.39 \pm 1.09$ | $85.42 \pm 0.75$ | $93.79 \pm 1.57$ | $\mathbf{91.15 \pm 2.10}$ | $\mathbf{85.18 \pm 1.08}$ |
| C = 2 | Per-FedAvg | $\mathbf{93.45 \pm 0.28}$ | $86.29 \pm 0.93$ | $87.38 \pm 1.19$ | $62.01 \pm 5.60$ | $86.28 \pm 1.32$ |
| | pFedMe | $73.16 \pm 0.45$ | $68.01 \pm 0.55$ | $69.76 \pm 0.68$ | $49.97 \pm 0.58$ | $60.16 \pm 1.50$ |
| | FedBC | $78.04 \pm 0.57$ | $73.24 \pm 0.47$ | $74.95 \pm 0.36$ | $53.83 \pm 0.56$ | $71.22 \pm 1.21$ |
| | FedBC -FineTune | $89.27 \pm 0.29$ | $77.64 \pm 0.28$ | $79.69 \pm 0.11$ | $56.83 \pm 0.34$ | $80.63 \pm 0.51$ |
| | Per-FedBC | $93.09 \pm 0.41$ | $\mathbf{86.55 \pm 1.76}$ | $\mathbf{88.47 \pm 2.11}$ | $\mathbf{70.20 \pm 4.60}$ | $\mathbf{87.47 \pm 1.12}$ |
| C = 3 | Per-FedAvg | $\mathbf{85.79 \pm 0.87}$ | $75.69 \pm 2.30$ | $89.14 \pm 0.71$ | $90.98 \pm 3.37$ | $81.63 \pm 3.32$ |
| | pFedMe | $57.93 \pm 1.05$ | $47.84 \pm 1.17$ | $70.16 \pm 0.60$ | $68.76 \pm 0.91$ | $59.71 \pm 0.50$ |
| | FedBC | $60.31 \pm 0.74$ | $52.77 \pm 1.34$ | $73.09 \pm 0.85$ | $74.56 \pm 0.90$ | $65.05 \pm 1.00$ |
| | FedBC -FineTune | $74.42 \pm 0.29$ | $67.46 \pm 0.59$ | $90.08 \pm 0.42$ | $88.84 \pm 0.82$ | $72.60 \pm 0.34$ |
| | Per-FedBC | $85.61 \pm 1.18$ | $\mathbf{80.96 \pm 2.50}$ | $\mathbf{91.22 \pm 0.95}$ | $\mathbf{91.85 \pm 1.54}$ | $\mathbf{83.40 \pm 1.41}$ |

communication, test accuracy for min and max users are nearly the same, as shown in Figure 4a. Whereas for FedAvg (Figure 4b), this gap can be as large as 100% even after nearly 200 rounds of communication. As compared to q-Fedavg (Figure 4c), FedProx (Figure 4d), or Scaffold (Figure 4e), `FedBC` has a much higher fraction of points at which test accuracy for min and max user overlap, which indicates that they are being treated equally well (*reducing performance disparity*). We also present additional results in Appendix E (Figure 10-12) for $E = 25, 50$, because the local model differs more from the global model as $E$ increases. The trend is similar, and FedBC can still make good predictions on the min user's local data despite an increase in the performance gap compared to $E = 5$. This is significantly different from the case of FedAvg, in which its model fails to make any correct predictions on the min user's local data for the majority of times, as shown in Figure 10-12 in Appendix E. In essence, `FedBC` has the best performance because it allows users to participate fairly in updating the global model. This performance benefit is credited to using non-zero $\gamma_i$, which is the main contribution of this work.

**Real Dataset Experiments:** The experiments on real datasets are in line with our previous observations in Figure 3. We first report the results for global model performance on the CIFAR-10 dataset in Table 2 for $E = 5$ (see Appendix 6 for $E = 1$). We note that FedBC outperforms FedAvg by 7.89% for $C = 3$. For the

most challenging situation of $C = 1$, FedBC outperforms all other baselines. The superior performance of FedBC is attributed to the fact that we observe unbiasedness in computing the coefficient for the global model when $C = 1, 2$, or 3 (see Figure 13 in Appendix F.1). We also observe the high uniformity in test accuracy over users' local data for all classes with different power law exponents (see Figure 15 in Appendix F.1).

We show the classification results on MNIST in Table 7 and Table 8 in Appendix F.2 and observe that FedBC outperforms all the other baselines. We also present the results on the Shakespeare dataset in Table 9 in Appendix F.3. From Table 2, we also notice the global performance of pFedMe and Per-FedAvg is worse than that of FedAvg, FedProx or FedBC when $C = 1$. This is mainly because personalized algorithms are designed to optimize the local objective of (2). However, this may create conflicts with the global objective in (1) and lead to poor global performance.

**Local Performance.** We have established that a non-zero $\gamma_i$ in FedBC leads to obtaining a better global model. This is because it provides additional freedom to automatically decide the contributions of local models rather than sticking to a uniform averaging, as done in existing FL methods. But a remark regarding the individual performance of local models $\mathbf{x}_i^t$ is due. We can evaluate the test accuracy of local model $\mathbf{x}_i$ at each device $i$ to see how it performs with respect to local test data. Table 3 presents the local performance of FedBC and other personalization algorithms. We observe that FedBC performs better and worse than pFedMe and Per-FedAvg, respectively. We can expect this performance because our algorithm is not designed to just focus on improving the local performance compared to pFedMe and Per-FedAvg. But an interesting point is that we can utilize the local models $\mathbf{x}_i^t$ obtained by FeBC to act as a good initializer, and after doing some fine tuning at device $i$, can improve the local performance as well. For instance, by doing 1-step fine-tuning on the test data (we call it FedBC-FineTune), we can improve the local performance of FedBC. For example, FedBC-FineTune achieves a 7.55% performance increase over FedBC when $C = 3$. To further improve the local model performance, as an additional experimental study, we incorporated MAML-type training into FedBC (cf. Algorithm 3 in Appendix D.4), which we call Per-FedBC algorithm, we can significantly improve the local performance for both $C = 1$ and $C = 3$.

**Takeaways:** In summary, we experimentally show that FedBC has the best global performance when compared against all baselines (addresses ①). Moreover, FedBC has reasonably good local performance but can be improved by fine-tuning or performing MAML-type training (addresses ②). We leave the question of how to fully exploit the advantage in the freedom of choosing $\gamma_i$ to achieve personalization for future work.

## 6 Conclusions

In this work, we delved into the intricate relationship between local and global model performance in federated learning (FL). We introduced a new proximity constraint to the FL framework, enabling the automatic determination of local model contributions to the global model. Our research demonstrates that by recognizing the flexibility to not force consensus among local models, we can simultaneously improve both the global and local performance of FL algorithms. Building on this insight, we developed the novel FedBC algorithm, which has been shown to perform well across a broad range of synthetic and real data sets. It outperforms state-of-the-art methods by automatically calibrating local and global models efficiently across devices.

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

# Appendix

## Table of Contents

## A  Preliminary Results

Before starting the analysis, let us revisit the definition of the Lagrangian and the gradients with respect to each variable given by

$$
\begin{aligned}
\mathcal{L}\left(\{\mathbf{x}\}_{\forall i}, \mathbf{z}, \{\lambda_i\}_{\forall i}\right) &= \sum_{i=1}^{N} \mathcal{L}_i(\mathbf{x}_i, \mathbf{z}, \lambda_i) \\
&= \sum_{i=1}^{N} \left[ f_i(\mathbf{x}_i) + \lambda_i \left( \|\mathbf{x}_i - \mathbf{z}\|^2 - \gamma_i \right) \right],
\end{aligned}
\tag{17}
$$

and

$$
\nabla_{\mathbf{x}_i} \mathcal{L}_i(\mathbf{x}_i, \mathbf{z}, \lambda_i) = \nabla_{\mathbf{x}_i} f_i(\mathbf{x}_i) + (2\lambda_i)(\mathbf{x}_i - \mathbf{z}),
\tag{18}
$$

$$
\nabla_{\mathbf{z}} \mathcal{L}_i(\mathbf{x}_i, \mathbf{z}, \lambda_i) = -\sum_{i=1}^{N} (2\lambda_i)(\mathbf{x}_i - \mathbf{z}),
\tag{19}
$$

$$
\nabla_{\lambda_i} \mathcal{L}_i(\mathbf{x}_i, \mathbf{z}, \lambda_i) = \|\mathbf{x}_i - \mathbf{z}\|^2 - \gamma_i.
\tag{20}
$$

**Lemma A.1.** *Under Assumption 4.1-4.2, for the proposed algorithm iterates, it holds that*

*(i) the average of the expected value of the gradient of Lagrangian with respect to primal variable satisfies*

$$
\frac{1}{N} \sum_{i=1}^{N} \mathbb{E}[\|\nabla_{\mathbf{x}_i} \mathcal{L}_i(\mathbf{x}_i^t, \mathbf{z}^t, \lambda_i^t)\|^2] \leq 2M^2 \left[ \frac{1}{N} \sum_{i=1}^{N} \mathbb{E}[\|\mathbf{x}_i^{t+1} - \mathbf{x}_i^t\|^2] + \epsilon \right],
\tag{21}
$$

*where $M_i := L_i + 2\lambda_{\max}$ and $M := \max_i M_i$.*

*(ii) Also, the first term on the right-hand side of* (21) *satisfies*

$$
\begin{aligned}
\frac{1}{N} \sum_{i=1}^{N} \mathbb{E}\left[\|\mathbf{x}_i^{t+1} - \mathbf{x}_i^t\|^2\right] \leq &\frac{1}{N(\beta-1)L_{\min}} \sum_{i=1}^{N} \mathbb{E}\left[\mathcal{L}_i(\mathbf{x}_i^t, \mathbf{z}^t, \lambda_i^t) - \mathcal{L}_i(\mathbf{x}_i^{t+1}, \mathbf{z}^{t+1}, \lambda_i^{t+1})\right] + \frac{1}{N(\beta-1)L_{\min}} \sum_{i=1}^{N} \frac{\epsilon}{2L_i}. \\
&+ \alpha G^2 + \frac{\lambda_{\max}}{N(\beta-1)L_{\min}} \sum_{i=1}^{N} \|\mathbf{x}_i^{t+1} - \mathbf{z}^{t+1}\|^2,
\end{aligned}
\tag{22}
$$

*where $\beta > 1$, $L_{\min} = \min_i L_i$.*

*Proof.* **Proof of Statement (i):** Let us consider the gradient of the Lagrangian with respect to the primal variable $\mathbf{x}_i$ for each agent $i$ which is $\nabla_{\mathbf{x}_i} \mathcal{L}_i(\mathbf{x}_i, \mathbf{z}, \lambda_i)$ (cf. 27) given by

$$
\|\nabla_{\mathbf{x}_i} \mathcal{L}_i(\mathbf{x}_i^t, \mathbf{z}^t, \lambda_i^t)\| = \|\nabla_{\mathbf{x}_i} f_i(\mathbf{x}_i^t) + (2\lambda_i^t)(\mathbf{x}_i^t - \mathbf{z}^t)\|.
\tag{23}
$$

Since we use GD oracle to solve the primal problem and obtain $\mathbf{x}_i^{t+1}$ for given $\mathbf{z}^t$ and $\lambda_i^t$ such that

$$
\|\nabla_{\mathbf{x}_i} \mathcal{L}_i(\mathbf{x}_i^{t+1}, \mathbf{z}^t, \lambda_i^t)\|^2 \leq \epsilon.
\tag{24}
$$

For a SGD oracle, the above condition would become

$$
\mathbb{E}\left[\|\nabla_{\mathbf{x}_i} \mathcal{L}_i(\mathbf{x}_i^{t+1}, \mathbf{z}^t, \lambda_i^t)\|^2\right] \leq \epsilon.
\tag{25}
$$

and would not change the analysis much, so we stick to the analysis for GD-based oracle. Now, let us define that

$$
\nabla f_i(\mathbf{x}_i^{t+1}) + (2\lambda_i^t)(\mathbf{x}_i^{t+1} - \mathbf{z}^t) = \mathbf{e}_i^{t+1},
\tag{26}
$$

where it holds that $\|\mathbf{e}_i^{t+1}\|^2 \leq \epsilon$ from (25). From (26), it holds that $\nabla f_i(\mathbf{x}_i^{t+1}) + (2\lambda_i^t)(\mathbf{x}_i^{t+1} - \mathbf{z}^t) - \mathbf{e}_i^{t+1} = 0$, we can write (23) as

$$
\begin{aligned}
\|\nabla_{\mathbf{x}_i}\mathcal{L}_i(\mathbf{x}_i^t, \mathbf{z}^t, \lambda_i^t)\| =& \|\nabla_{\mathbf{x}_i}f_i(\mathbf{x}_i^t) + (2\lambda_i^t)(\mathbf{x}_i^t - \mathbf{z}^t) - \nabla f_i(\mathbf{x}_i^{t+1}) - (2\lambda_i^t)(\mathbf{x}_i^{t+1} - \mathbf{z}^t) + \mathbf{e}_i^{t+1}\| \\
=& \|\nabla_{\mathbf{x}_i}f_i(\mathbf{x}_i^t) - \nabla f_i(\mathbf{x}_i^{t+1}) + (2\lambda_i^t)(\mathbf{x}_i^t - \mathbf{x}_i^{t+1}) + \mathbf{e}_i^{t+1}\| \\
\leq& L_i\|\mathbf{x}_i^{t+1} - \mathbf{x}_i^t\| + (2\lambda_i^t)\|\mathbf{x}_i^{t+1} - \mathbf{x}_i^t\| + \|\mathbf{e}_i^{t+1}\| \\
\leq& (L_i + 2\lambda_{\max})\|\mathbf{x}_i^{t+1} - \mathbf{x}_i^t\| + \sqrt{\epsilon},
\end{aligned}
\tag{27}
$$

where we utilize triangle inequality. We use the upper bound for each term in the last inequality to obtain (27). Next, let us define $M_i := L_i + 2\lambda_{\max}$ and taking the square on both sides in (27), we get

$$
\|\nabla_{\mathbf{x}_i}\mathcal{L}_i(\mathbf{x}_i^t, \mathbf{z}^t, \lambda_i^t)\|^2 \leq 2(M)^2\|\mathbf{x}_i^{t+1} - \mathbf{x}_i^t\|^2 + 2\epsilon,
\tag{28}
$$

where $M := \max_i M_i$. Take expectation on both sides in (28), then take summation over $i$, we can write

$$
\frac{1}{N}\sum_{i=1}^N \mathbb{E}[\|\nabla_{\mathbf{x}_i}\mathcal{L}_i(\mathbf{x}_i^t, \mathbf{z}^t, \lambda_i^t)\|^2] \leq 2M^2\left[\frac{1}{N}\sum_{i=1}^N \mathbb{E}[\|\mathbf{x}_i^{t+1} - \mathbf{x}_i^t\|^2] + \epsilon\right].
\tag{29}
$$

Hence proved.

**Proof of Statement (ii):** In order to bound the right-hand side of (29), let us consider the Lagrangian difference as

$$
\begin{aligned}
\mathcal{L}_i(\mathbf{x}_i^{t+1}, \mathbf{z}^{t+1}, \lambda_i^{t+1}) - \mathcal{L}_i(\mathbf{x}_i^t, \mathbf{z}^t, \lambda_i^t) =& \underbrace{\mathcal{L}_i(\mathbf{x}_i^{t+1}, \mathbf{z}^t, \lambda_i^t) - \mathcal{L}_i(\mathbf{x}_i^t, \mathbf{z}^t, \lambda_i^t)}_{I} \\
& \underbrace{\mathcal{L}_i(\mathbf{x}_i^{t+1}, \mathbf{z}^t, \lambda_i^{t+1}) - \mathcal{L}_i(\mathbf{x}_i^{t+1}, \mathbf{z}^t, \lambda_i^t)}_{II} \\
& \underbrace{\mathcal{L}_i(\mathbf{x}_i^{t+1}, \mathbf{z}^{t+1}, \lambda_i^{t+1}) - \mathcal{L}_i(\mathbf{x}_i^{t+1}, \mathbf{z}^t, \lambda_i^{t+1})}_{III}.
\end{aligned}
\tag{30}
$$

Next, we establish upper bounds for the terms $I, II$, and $III$ separately as follows.

**Bound on $I$:** Using the definition of the Lagrangian, we can write

$$
\begin{aligned}
I =& f_i(\mathbf{x}_i^{t+1}) + \lambda_i^t\left(\|\mathbf{x}_i^{t+1} - \mathbf{z}^t\|^2 - \gamma_i\right) - f_i(\mathbf{x}_i^t) - \lambda_i^t\left(\|\mathbf{x}_i^t - \mathbf{z}^t\|^2 - \gamma_i\right) \\
=& f_i(\mathbf{x}_i^{t+1}) - f_i(\mathbf{x}_i^t) + \lambda_i^t\left(\|\mathbf{x}_i^{t+1} - \mathbf{z}^t\|^2 - \|\mathbf{x}_i^t - \mathbf{z}^t\|^2\right).
\end{aligned}
\tag{31}
$$

From the Lipschitz smoothness of the local objective function (cf. Assumption 4.2), we can write

$$
I = \langle\nabla f_i(\mathbf{x}_i^{t+1}), \mathbf{x}_i^{t+1} - \mathbf{x}_i^t\rangle + \frac{L_i}{2}\|\mathbf{x}_i^{t+1} - \mathbf{x}_i^t\|^2 + (\lambda_i^t)\left(\|\mathbf{x}_i^{t+1} - \mathbf{z}^t\|^2 - \|\mathbf{x}_i^t - \mathbf{z}^t\|^2\right).
\tag{32}
$$

Utilizing the equality $\|\mathbf{a}\|^2 - \|\mathbf{b}\|^2 = \langle\mathbf{a} + \mathbf{b}, \mathbf{a} - \mathbf{b}\rangle$, we obtain

$$
\begin{aligned}
I =& \langle\nabla f_i(\mathbf{x}_i^{t+1}), \mathbf{x}_i^{t+1} - \mathbf{x}_i^t\rangle + \frac{L_i}{2}\|\mathbf{x}_i^{t+1} - \mathbf{x}_i^t\|^2 + (\lambda_i^t)\left(\langle 2\mathbf{x}_i^{t+1} - \mathbf{x}_i^{t+1} + \mathbf{x}_i^t - 2\mathbf{z}^t, \mathbf{x}_i^{t+1} - \mathbf{x}_i^t\rangle\right) \\
=& \langle\nabla f_i(\mathbf{x}_i^{t+1}), \mathbf{x}_i^{t+1} - \mathbf{x}_i^t\rangle + \frac{L_i}{2}\|\mathbf{x}_i^{t+1} - \mathbf{x}_i^t\|^2 + 2(\lambda_i^t)\left(\langle\mathbf{x}_i^{t+1} - \mathbf{z}^t, \mathbf{x}_i^{t+1} - \mathbf{x}_i^t\rangle\right) - (\lambda_i^t)\|\mathbf{x}_i^{t+1} - \mathbf{x}_i^t\|^2.
\end{aligned}
\tag{33}
$$

After rearranging the terms, we get

$$
I = \langle\nabla f_i(\mathbf{x}_i^{t+1}) + 2(\lambda_i^t)(\mathbf{x}_i^{t+1} - \mathbf{z}^t), \mathbf{x}_i^{t+1} - \mathbf{x}_i^t\rangle + \frac{L_i}{2}\|\mathbf{x}_i^{t+1} - \mathbf{x}_i^t\|^2 - (\lambda_i^t)\|\mathbf{x}_i^{t+1} - \mathbf{x}_i^t\|^2.
\tag{34}
$$

Using the Peter-Paul inequality, we obtain

$$
I = \frac{1}{2L_i}\|\nabla f_i(\mathbf{x}_i^{t+1}) + 2\lambda_i^t(\mathbf{x}_i^{t+1} - \mathbf{z}^t)\|^2 + L_i\|\mathbf{x}_i^{t+1} - \mathbf{x}_i^t\|^2 - (\lambda_i^t)\|\mathbf{x}_i^{t+1} - \mathbf{x}_i^t\|^2.
\tag{35}
$$

From the optimality condition of the local oracle, we know that $\nabla f_i(\mathbf{x}_i^{t+1}) + 2\lambda_i^t(\mathbf{x}_i^{t+1} - \mathbf{z}^t) = \mathbf{e}_i^{t+1}$, we can write

$$I \leq \frac{\epsilon}{2L_i} - \left(\lambda_i^t - L_i\right) \|\mathbf{x}_i^{t+1} - \mathbf{x}_i^t\|^2. \tag{36}$$

We obtain a condition on $\lambda_i^t \geq L_i$ which results in the lower bound for the dual variable. It is telling us that we care more about devices for which the local objective is smooth than the others with non-smooth local objectives. The lower the $L_i$ is, the lower we care about the consensus for that particular device.

**Bound on $II$:** Let us consider the term $II$ after substituting the value of the Lagrangian, we get

$$\begin{aligned}
II =& (\lambda_i^{t+1} - \lambda_i^t) \left(\|\mathbf{x}_i^{t+1} - \mathbf{z}^t\|^2 - \gamma_i\right). \\
\leq& |\lambda_i^{t+1} - \lambda_i^t| \cdot |\left(\|\mathbf{x}_i^{t+1} - \mathbf{z}^t\|^2 - \gamma_i\right)|
\end{aligned} \tag{37}$$

From the update of the dual variable, we can write

$$\begin{aligned}
II \leq \alpha \left(\|\mathbf{x}_i^{t+1} - \mathbf{z}^t\|^2 - \gamma_i\right)^2 \leq& \alpha\|\mathbf{x}_i^{t+1} - \mathbf{z}^t\|^4 + \alpha\gamma_i^2 \\
\leq& 16\alpha R^4 + \alpha\gamma_i^2 \\
=& \alpha G^2,
\end{aligned} \tag{38}$$

where $G := \sqrt{16R^4 + \gamma_i^2}$, which follows from Assumption 4.1.

**Bound on $III$:** Let us consider the term $III$ after substituting the value of the Lagrangian, we get

$$\begin{aligned}
III = \left(\lambda_i^{t+1}\right) \left(\|\mathbf{x}_i^{t+1} - \mathbf{z}^{t+1}\|^2 - \|\mathbf{x}_i^{t+1} - \mathbf{z}^t\|^2\right) \\
\leq (\lambda_{\max}) \|\mathbf{x}_i^{t+1} - \mathbf{z}^{t+1}\|^2.
\end{aligned} \tag{39}$$

Next, we utilize the upper bounds on $I, II$, and $III$ into (30) to obtain

$$\begin{aligned}
\mathcal{L}_i(\mathbf{x}_i^{t+1}, \mathbf{z}^{t+1}, \lambda_i^{t+1}) - \mathcal{L}_i(\mathbf{x}_i^t, \mathbf{z}^t, \lambda_i^t) \leq& \frac{\epsilon}{2L_i} - \left(\lambda_i^t - L_i\right) \|\mathbf{x}_i^{t+1} - \mathbf{x}_i^t\|^2 + \alpha G^2 \\
&+ (\lambda_{\max}) \|\mathbf{x}_i^{t+1} - \mathbf{z}^{t+1}\|^2.
\end{aligned} \tag{40}$$

Due to the device sampling happening in our proposed algorithm, the above inequality holds for $i \in \mathcal{S}_t$ only. Hence, let us take the summation over $i \in \mathcal{S}_t$ as follows

$$\sum_{i \in \mathcal{S}_t} [\mathcal{L}_i(\mathbf{x}_i^{t+1}, \mathbf{z}^{t+1}, \lambda_i^{t+1}) - \mathcal{L}_i(\mathbf{x}_i^t, \mathbf{z}^t, \lambda_i^t)] \leq \sum_{i \in \mathcal{S}_t} \left[\frac{\epsilon}{2L_i} - \left(\lambda_i^t - L_i\right) \|\mathbf{x}_i^{t+1} - \mathbf{x}_i^t\|^2 + \alpha G^2 \right.$$

$$\left. + (\lambda_{\max}) \|\mathbf{x}_i^{t+1} - \mathbf{z}^{t+1}\|^2\right]. \tag{41}$$

We note that it holds from the definition of indicator function that

$$\sum_{i=1}^N \mathbb{I}_{\{i \in \mathcal{S}_t\}} \cdot [\mathcal{L}_i(\mathbf{x}_i^{t+1}, \mathbf{z}^{t+1}, \lambda_i^{t+1}) - \mathcal{L}_i(\mathbf{x}_i^t, \mathbf{z}^t, \lambda_i^t)] \leq \sum_{i=1}^N \mathbb{I}_{\{i \in \mathcal{S}_t\}} \cdot \left[\frac{\epsilon}{2L_i} - \left(\lambda_i^t - L_i\right) \|\mathbf{x}_i^{t+1} - \mathbf{x}_i^t\|^2 + \alpha G^2 \right.$$

$$\left. + (\lambda_{\max}) \|\mathbf{x}_i^{t+1} - \mathbf{z}^{t+1}\|^2\right]. \tag{42}$$

Therefore, before moving forward, we calculate the conditional expectation $\mathbb{E}_{\mathcal{S}_t}$ on both sides in the above expression to remove the randomness due to device sampling. Hence, after taking expectation with respect to $\mathcal{S}_t$ which we denote by $\mathbb{E}_{\mathcal{S}_t}$, it holds that

$$\sum_{i=1}^N \mathbb{E}_{\mathcal{S}_t} \left[\mathbb{I}_{\{i \in \mathcal{S}_t\}}\right] \cdot [\mathcal{L}_i(\mathbf{x}_i^{t+1}, \mathbf{z}^{t+1}, \lambda_i^{t+1}) - \mathcal{L}_i(\mathbf{x}_i^t, \mathbf{z}^t, \lambda_i^t)] \leq \sum_{i=1}^N \mathbb{E}_{\mathcal{S}_t} \left[\mathbb{I}_{\{i \in \mathcal{S}_t\}}\right] \cdot \left[\frac{\epsilon}{2L_i} - \left(\lambda_i^t - L_i\right) \|\mathbf{x}_i^{t+1} - \mathbf{x}_i^t\|^2 + \alpha G^2 \right.$$

$$\left. + (\lambda_{\max}) \|\mathbf{x}_i^{t+1} - \mathbf{z}^{t+1}\|^2\right]. \tag{43}$$

We note that $\mathbb{E}_{\mathcal{S}_t}\left[\mathbb{I}_{\{j \in \mathcal{S}_t\}}\right] = \mathbb{P}(j \in \mathcal{S}_t) = \frac{S}{N}$, since we are sampling $S$ devices uniformly from $N$. Utilizing this in the right hand side of (43) and taking total expectation, we get

$$\frac{S}{N}\sum_{i=1}^{N}\mathbb{E}[\mathcal{L}_i(\mathbf{x}_i^{t+1}, \mathbf{z}^{t+1}, \lambda_i^{t+1}) - \mathcal{L}_i(\mathbf{x}_i^t, \mathbf{z}^t, \lambda_i^t)] \leq \frac{\epsilon S}{2N}\sum_{i=1}^{N}\frac{1}{L_i} - \frac{S}{N}\sum_{i=1}^{N}\mathbb{E}\left(\left(\lambda_i^t - L_i\right)\|\mathbf{x}_i^{t+1} - \mathbf{x}_i^t\|^2\right) + \alpha S G^2$$
$$+ (\lambda_{\max})\frac{S}{N}\sum_{i=1}^{N}\mathbb{E}\|\mathbf{x}_i^{t+1} - \mathbf{z}^{t+1}\|^2. \tag{44}$$

After rearranging the terms, we can write

$$\frac{1}{N}\sum_{i=1}^{N}\left(\lambda_i^t - L_i\right)\mathbb{E}\left[\|\mathbf{x}_i^{t+1} - \mathbf{x}_i^t\|^2\right] \leq \frac{1}{N}\sum_{i=1}^{N}\mathbb{E}\left[\mathcal{L}_i(\mathbf{x}_i^t, \mathbf{z}^t, \lambda_i^t) - \mathcal{L}_i(\mathbf{x}_i^{t+1}, \mathbf{z}^{t+1}, \lambda_i^{t+1})\right] + \frac{1}{N}\sum_{i=1}^{N}\frac{\epsilon}{2L_i} + \alpha G^2.$$
$$+ (\lambda_{\max})\frac{1}{N}\sum_{i=1}^{N}\mathbb{E}\|\mathbf{x}_i^{t+1} - \mathbf{z}^{t+1}\|^2. \tag{45}$$

Under the condition that $\lambda_i^t \geq \beta L_i$ for all $i$ with $\beta > 1$, it holds $\lambda_i^t - L_i \geq (\beta - 1)L_i \geq (\beta - 1)L_{\min}$ where $L_{\min} := \min_i L_i$, then we have

$$\frac{1}{N}\sum_{i=1}^{N}\mathbb{E}\left[\|\mathbf{x}_i^{t+1} - \mathbf{x}_i^t\|^2\right] \leq \frac{1}{N(\beta - 1)L_{\min}}\sum_{i=1}^{N}\mathbb{E}\left[\mathcal{L}_i(\mathbf{x}_i^t, \mathbf{z}^t, \lambda_i^t) - \mathcal{L}_i(\mathbf{x}_i^{t+1}, \mathbf{z}^{t+1}, \lambda_i^{t+1})\right] + \frac{1}{N(\beta - 1)L_{\min}}\sum_{i=1}^{N}\frac{\epsilon}{2L_i}.$$
$$+ \alpha G^2 + \frac{\lambda_{\max}}{N(\beta - 1)L_{\min}}\sum_{i=1}^{N}\|\mathbf{x}_i^{t+1} - \mathbf{z}^{t+1}\|^2. \tag{46}$$

Hence proved. □

**Lemma A.2.** *Under Assumption 4.1-4.3, for the iterates of proposed Algorithm 1, we establish that the consensus error satisfies:*

$$\frac{1}{T}\sum_{t=0}^{T-1}\left[\frac{1}{N}\sum_{i=1}^{N}\mathbb{E}\left[\|\mathbf{x}_i^{t+1} - \mathbf{z}^{t+1}\|^2\right]\right] \leq \frac{\epsilon S(N-1)}{C_0^2 N^2}\sum_{i=1}^{N}\frac{1}{(\beta L_i)^2} + \frac{\delta^2 S(N-1)}{C_0^2 N^2}\sum_{i=1}^{N}\frac{1}{(\beta L_i)^4}, \tag{47}$$

*where $\epsilon$ is the accuracy with each agent solving the local optimization problem, $\delta$ is the heterogeneity parameter (cf. Assumption 4.3), $C_0 := \left(1 - \frac{\lambda_{\max}}{4L}\right)$, $M_i = L_i + 2\lambda_{\max}$, $M := \max_i M_i$, $\lambda_{\max}$ is the dual variable upper bound, $L_i$ is the Lipschitz smoothness parameter of device $i$, $N$ is the number of devices, and we sample $S$ devices uniformly from $N$ devices.*

*Proof.* From the server update (cf Algorithm 1), we note that

$$\|\mathbf{x}_i^{t+1} - \mathbf{z}^{t+1}\|^2 = \left\|\mathbf{x}_i^{t+1} - \frac{1}{\sum_{j \in \mathcal{S}_t}\lambda_j^{t+1}}\sum_{j \in \mathcal{S}_t}\lambda_j^{t+1}\mathbf{x}_j^{t+1}\right\|^2$$
$$= \left\|\frac{1}{\sum_{j \in \mathcal{S}_t}\lambda_j^{t+1}}\sum_{j \in \mathcal{S}_t}\lambda_j^{t+1}(\mathbf{x}_i^{t+1} - \mathbf{x}_j^{t+1})\right\|^2, \tag{48}$$

which holds true because $\mathbf{x}_i^{t+1}$ does not depend upon $j$ index. Next, pulling the summation outside the norm because it's a convex function via Jensen's inequality, we can write

$$\|\mathbf{x}_i^{t+1} - \mathbf{z}^{t+1}\|^2 \leq \sum_{j \in \mathcal{S}_t}\frac{\lambda_j^{t+1}}{\sum_{j \in \mathcal{S}_t}\lambda_j^{t+1}}\|\mathbf{x}_i^{t+1} - \mathbf{x}_j^{t+1}\|^2 \tag{49}$$
$$\leq \sum_{j \in \mathcal{S}_t}\|\mathbf{x}_i^{t+1} - \mathbf{x}_j^{t+1}\|^2, \tag{50}$$

where the second inequality holds due to the fact that $\frac{\lambda_j^{t+1}}{\sum_{j \in \mathcal{S}_t} \lambda_j^{t+1}} \leq 1$. Now we calculate the expectation of the device sampling given by $\mathbb{E}_{\mathcal{S}_t}$ to obtain

$$\mathbb{E}_{\mathcal{S}_t}\left[\|\mathbf{x}_i^{t+1} - \mathbf{z}^{t+1}\|^2\right] \leq \mathbb{E}_{\mathcal{S}_t}\left[\sum_{j \in \mathcal{S}_t} \|\mathbf{x}_i^{t+1} - \mathbf{x}_j^{t+1}\|^2\right] \tag{51}$$

$$= \mathbb{E}_{\mathcal{S}_t}\left[\sum_{j=1}^{N} \|\mathbf{x}_i^{t+1} - \mathbf{x}_j^{t+1}\|^2 \cdot \mathbb{I}_{\{j \in \mathcal{S}_t\}}\right] \tag{52}$$

$$= \sum_{j=1}^{N} \|\mathbf{x}_i^{t+1} - \mathbf{x}_j^{t+1}\|^2 \cdot \mathbb{E}_{\mathcal{S}_t}\left[\mathbb{I}_{\{j \in \mathcal{S}_t\}}\right]. \tag{53}$$

We note that $\mathbb{E}_{\mathcal{S}_t}\left[\mathbb{I}_{\{j \in \mathcal{S}_t\}}\right] = \mathbb{P}(j \in \mathcal{S}_t)$. Since we sample $S$ devices uniformly from $N$ devices, it holds that $\mathbb{P}(j \in \mathcal{S}_t) = \frac{S}{N}$. Utilizing this in the right hand side of (53), we get

$$\mathbb{E}_{\mathcal{S}_t}\left[\|\mathbf{x}_i^{t+1} - \mathbf{z}^{t+1}\|^2\right] \leq \sum_{j=1}^{N} \|\mathbf{x}_i^{t+1} - \mathbf{x}_j^{t+1}\|^2 \cdot \mathbb{P}(j \in \mathcal{S}_t) \tag{54}$$

$$= \frac{S}{N} \sum_{j=1}^{N} \|\mathbf{x}_i^{t+1} - \mathbf{x}_j^{t+1}\|^2 \tag{55}$$

$$= \frac{S}{N} \sum_{j \neq i} \|\mathbf{x}_i^{t+1} - \mathbf{x}_j^{t+1}\|^2$$

$$\leq \frac{S(N-1)}{N} \triangle \mathbf{x}^{t+1}, \tag{56}$$

where we define $\triangle \mathbf{x}^{t+1} := \max_{i,j} \|\mathbf{x}_i^{t+1} - \mathbf{x}_j^{t+1}\|^2$. We start with bounding the difference $\|\mathbf{x}_i^{t+1} - \mathbf{x}_j^{t+1}\|$ for any arbitrary $i \neq j$. Before starting, we recall that

$$\nabla f_i(\mathbf{x}_i^{t+1}) + \left(2\lambda_i^t\right)\left(\mathbf{x}_i^{t+1} - \mathbf{z}^t\right) = \mathbf{e}_i^{t+1}, \tag{57}$$

which implies that

$$\mathbf{x}_i^{t+1} = \mathbf{z}^t + \frac{\mathbf{e}_i^{t+1} - \nabla f_i(\mathbf{x}_i^{t+1})}{\lambda_i^t}. \tag{58}$$

From the above expression, we can write

$$\|\mathbf{x}_i^{t+1} - \mathbf{x}_j^{t+1}\| = \left\|\mathbf{z}^t + \frac{\mathbf{e}_i^{t+1} - \nabla f_i(\mathbf{x}_i^{t+1})}{a_i} - \mathbf{z}^t + \frac{\mathbf{e}_j^{t+1} - \nabla f_j(\mathbf{x}_j^{t+1})}{a_j}\right\|, \tag{59}$$

where $a_i := (\lambda_i^t)$. After rearranging the terms and applying [triangle inequality](triangle inequality), we can write

$$\|\mathbf{x}_i^{t+1} - \mathbf{x}_j^{t+1}\| \leq \left\|\frac{\mathbf{e}_i^{t+1} - \nabla f_i(\mathbf{x}_i^{t+1})}{a_i} - \frac{\mathbf{e}_j^{t+1} - \nabla f_j(\mathbf{x}_j^{t+1})}{a_j}\right\|$$

$$= \frac{1}{a_i a_j}\|a_j \mathbf{e}_i^{t+1} - a_i \mathbf{e}_j^{t+1} - a_j \nabla f_i(\mathbf{x}_i^{t+1}) + a_i \nabla f_j(\mathbf{x}_j^{t+1})\|$$

$$\leq \frac{1}{a_i}\|\mathbf{e}_i^{t+1}\| + \frac{1}{a_j}\|\mathbf{e}_j^{t+1}\| + \frac{1}{a_i a_j}\|a_j \nabla f_i(\mathbf{x}_i^{t+1}) - a_i \nabla f_j(\mathbf{x}_j^{t+1})\|. \tag{60}$$

Next, note that since $a_i \geq \beta L_i$, which implies that $1/a_i \leq 1/(\beta L_i)$, hence we can write

$$\|\mathbf{x}_i^{t+1} - \mathbf{x}_j^{t+1}\| \leq \frac{1}{\beta L_i}\sqrt{\epsilon} + \frac{1}{(\beta L_i)^2}\|a_j \nabla f_i(\mathbf{x}_i^{t+1}) - a_i \nabla f_j(\mathbf{x}_j^{t+1})\|. \tag{61}$$

Consider the term $\|a_j \nabla f_i(\mathbf{x}_i^{t+1}) + a_i \nabla f_j(\mathbf{x}_j^{t+1})\|$ and we add subtract $\nabla f_i(\mathbf{x}_j^{t+1})$ as follows

$$
\begin{aligned}
\|a_j \nabla f_i(\mathbf{x}_i^{t+1}) + a_i \nabla f_j(\mathbf{x}_j^{t+1})\| &= \|a_j(\nabla f_i(\mathbf{x}_i^{t+1}) - \nabla f_i(\mathbf{x}_j^{t+1}) + \nabla f_i(\mathbf{x}_j^{t+1})) - a_i \nabla f_j(\mathbf{x}_j^{t+1})\| \\
&\leq a_j \|\nabla f_i(\mathbf{x}_i^{t+1}) - \nabla f_i(\mathbf{x}_j^{t+1})\| + \|a_j \nabla f_i(\mathbf{x}_j^{t+1}) - a_i \nabla f_j(\mathbf{x}_j^{t+1})\| \\
&\leq a_j L_i \|\mathbf{x}_i^{t+1} - \mathbf{x}_j^{t+1}\| + \delta
\end{aligned}
\tag{62}
$$

Using (62) into (61) and the fact that $a_i \leq \lambda_{\max}$. , we get

$$
\|\mathbf{x}_i^{t+1} - \mathbf{x}_j^{t+1}\| \leq \frac{1}{\beta L_i}\sqrt{\epsilon} + \frac{\lambda_{\max}}{(\beta)^2 L_i}\|\mathbf{x}_i^{t+1} - \mathbf{x}_j^{t+1}\| + \frac{1}{(\beta L_i)^2}\delta.
\tag{63}
$$

After rearranging the terms, we get

$$
\left(1 - \frac{\lambda_{\max}}{(\beta)^2 L_i}\right)\|\mathbf{x}_i^{t+1} - \mathbf{x}_j^{t+1}\| \leq \frac{1}{\beta L_i}\sqrt{\epsilon} + \frac{1}{(\beta L_i)^2}\delta.
\tag{64}
$$

The term $C_0 := \left(1 - \frac{\lambda_{\max}}{4 L_{\min}}\right)$ is strictly positive if $\lambda_{\max} < \beta^2 L_{\min}$ (this is the condition for $\lambda_{\max}$). Hence, we could divide both sides by $C_0$ to obtain

$$
\|\mathbf{x}_i^{t+1} - \mathbf{x}_j^{t+1}\| \leq \frac{1}{C_0(\beta L_i)}\sqrt{\epsilon} + \frac{1}{C_0(\beta L_i)^2}\delta.
\tag{65}
$$

Since the above bound holds for any $i, j$, it would also hold for the maximum, therefore we can write the upper bound in (56) as

$$
\begin{aligned}
\mathbb{E}_t\left[\|\mathbf{x}_i^{t+1} - \mathbf{z}^{t+1}\|^2\right] &\leq \frac{S(N-1)}{N}\max_{i,j}\|\mathbf{x}_i^{t+1} - \mathbf{x}_j^{t+1}\|^2 \\
&\leq \sum_{j \neq i} \frac{2S(N-1)}{C_0^2 N(\beta L_i)^2}\epsilon + \frac{S(N-1)}{C_0^2 N(\beta L_i)^4}\delta^2.
\end{aligned}
\tag{66}
$$

Take the average over all the agents, and then sum over $t$ on both sides to obtain

$$
\frac{1}{N}\sum_{t=0}^{T-1}\sum_{i=1}^{N}\mathbb{E}_t\left[\|\mathbf{x}_i^{t+1} - \mathbf{z}^{t+1}\|^2\right] \leq T\frac{\epsilon S(N-1)}{C_0^2 N^2}\sum_{i=1}^{N}\frac{1}{(\beta L_i)^2} + T\frac{\delta^2 S(N-1)}{C_0^2 N^2}\sum_{i=1}^{N}\frac{1}{(\beta L_i)^4}.
\tag{67}
$$

Taking total expectations on both sides, we get

$$
\frac{1}{N}\sum_{t=0}^{T-1}\sum_{i=1}^{N}\mathbb{E}\left[\|\mathbf{x}_i^{t+1} - \mathbf{z}^{t+1}\|^2\right] \leq T\frac{\epsilon S(N-1)}{C_0^2 N^2}\sum_{i=1}^{N}\frac{1}{(\beta L_i)^2} + T\frac{\delta^2 S(N-1)}{C_0^2 N^2}\sum_{i=1}^{N}\frac{1}{(\beta L_i)^4}.
\tag{68}
$$

Hence proved. $\qquad\square$

## B  Proof of Theorem 4.5

*Proof.* The goal here is to show that the term $\frac{1}{T}\sum_{t=0}^{T-1}\mathbb{E}\left[\|\nabla f(\mathbf{z}^t)\|^2\right]$ decreases as $T$ increases. In order to start the analysis, let us consider $\|\nabla f(\mathbf{z}^t)\|^2$ and write

$$
\|\nabla f(\mathbf{z}^t)\|^2 \leq \left\|\frac{1}{N}\sum_{i=1}^{N}\nabla f_i(\mathbf{z}^t) - \frac{1}{N}\sum_{i=1}^{N}\nabla_{\mathbf{x}_i}\mathcal{L}_i(\mathbf{x}_i^t, \mathbf{z}^t, \lambda_i^t) + \frac{1}{N}\sum_{i=1}^{N}\nabla_{\mathbf{x}_i}\mathcal{L}_i(\mathbf{x}_i^t, \mathbf{z}^t, \lambda_i^t)\right\|^2,
\tag{69}
$$

where we utilize the definition of gradient $\nabla f(\mathbf{z}^t) = \frac{1}{N}\sum_{i=1}^{N}\nabla f_i(\mathbf{z}^t)$ and add subtract the term $\frac{1}{N}\sum_{i=1}^{N}\nabla_{\mathbf{x}_i}\mathcal{L}_i(\mathbf{x}_i^t, \mathbf{z}^t, \lambda_i^t)$ inside the norm. Next, we utilize the upper bound $(a+b)^2 \leq 2a^2 + 2b^2$ and

obtain

$$\|\nabla f(\mathbf{z}^t)\|^2 \leq 2 \left\|\frac{1}{N}\sum_{i=1}^N \left[\nabla f_i(\mathbf{z}^t) - \nabla_{\mathbf{x}_i}\mathcal{L}_i(\mathbf{x}_i^t, \mathbf{z}^t, \lambda_i^t)\right]\right\|^2 + 2\left\|\frac{1}{N}\sum_{i=1}^N \nabla_{\mathbf{x}_i}\mathcal{L}_i(\mathbf{x}_i^t, \mathbf{z}^t, \lambda_i^t)\right\|^2, \tag{70}$$

$$\leq \frac{2}{N}\sum_{i=1}^N \left\|\nabla f_i(\mathbf{z}^t) - \nabla_{\mathbf{x}_i}\mathcal{L}_i(\mathbf{x}_i^t, \mathbf{z}^t, \lambda_i^t)\right\|^2 + \frac{2}{N}\sum_{i=1}^N \left\|\nabla_{\mathbf{x}_i}\mathcal{L}_i(\mathbf{x}_i^t, \mathbf{z}^t, \lambda_i^t)\right\|^2. \tag{71}$$

In (71), the inequality holds due to Jensen's inequality and we push $\frac{1}{N}\sum_{i=1}^n$ outside the norm to obtain the upper bound. Next, we substitute the definition of gradient provided in (18) to obtain

$$\|\nabla f(\mathbf{z}^t)\|^2 \leq \frac{2}{N}\sum_{i=1}^N \left\|\nabla f_i(\mathbf{z}^t) - \nabla_{\mathbf{x}_i} f_i(\mathbf{x}_i^t) - (2\lambda_i^t)\left(\mathbf{x}_i^t - \mathbf{z}^t\right)\right\|^2 + \frac{2}{N}\sum_{i=1}^N \left\|\nabla_{\mathbf{x}_i}\mathcal{L}_i(\mathbf{x}_i^t, \mathbf{z}^t, \lambda_i^t)\right\|^2,$$

$$\leq \frac{4}{N}\sum_{i=1}^N \left\|\nabla f_i(\mathbf{z}^t) - \nabla_{\mathbf{x}_i} f_i(\mathbf{x}_i^t)\right\|^2 + \frac{8\lambda_{\max}}{N}\sum_{i=1}^N \|\mathbf{x}_i^t - \mathbf{z}^t\|^2 + \frac{2}{N}\sum_{i=1}^N \left\|\nabla_{\mathbf{x}_i}\mathcal{L}_i(\mathbf{x}_i^t, \mathbf{z}^t, \lambda_i^t)\right\|^2. \tag{72}$$

where again we utilize $(a+b)^2 \leq 2a^2 + 2b^2$ and the compactness of the dual set $\Lambda$ to obtain the second inequality. From the Lipschitz continuous gradient of the local objectives (Assumption 4.2), we can upper bound the right-hand side of (72) as

$$\|\nabla f(\mathbf{z}^t)\|^2 \leq \frac{4}{N}\sum_{i=1}^N L_i^2\|\mathbf{z}^t - \mathbf{x}_i^t\|^2 + \frac{8\lambda_{\max}}{N}\sum_{i=1}^N \|\mathbf{x}_i^t - \mathbf{z}^t\|^2 + \frac{2}{N}\sum_{i=1}^N \left\|\nabla_{\mathbf{x}_i}\mathcal{L}_i(\mathbf{x}_i^t, \mathbf{z}^t, \lambda_i^t)\right\|^2$$

$$\leq \frac{4(L^2 + 2\lambda_{\max})}{N}\sum_{i=1}^N \|\mathbf{z}^t - \mathbf{x}_i^t\|^2 + \frac{2}{N}\sum_{i=1}^N \left\|\nabla_{\mathbf{x}_i}\mathcal{L}_i(\mathbf{x}_i^t, \mathbf{z}^t, \lambda_i^t)\right\|^2, \tag{73}$$

where $L = \max_i L_i$. Let us define $M := 4(L^2 + 2\lambda_{\max})$ and taking the expectation on both sides, we get

$$\mathbb{E}[\|\nabla f(\mathbf{z}^t)\|^2] \leq \frac{M}{N}\sum_{i=1}^N \mathbb{E}[\|\mathbf{z}^t - \mathbf{x}_i^t\|^2] + \frac{2}{N}\sum_{i=1}^N \mathbb{E}\left[\left\|\nabla_{\mathbf{x}_i}\mathcal{L}_i(\mathbf{x}_i^t, \mathbf{z}^t, \lambda_i^t)\right\|^2\right]. \tag{74}$$

Next, substitute the upper bound from (46) into the (29) after taking expectation of , we can write

$$\frac{1}{N}\sum_{i=1}^N \mathbb{E}[\|\nabla_{\mathbf{x}_i}\mathcal{L}_i(\mathbf{x}_i^t, \mathbf{z}^t, \lambda_i^t)\|^2] \leq 2M^2\left[\frac{1}{N}\sum_{i=1}^N \mathbb{E}[\|\mathbf{x}_i^{t+1} - \mathbf{x}_i^t\|^2] + \epsilon\right]$$

$$\leq \frac{2M^2}{N}\sum_{i=1}^N \mathbb{E}\left[\mathcal{L}_i(\mathbf{x}_i^t, \mathbf{z}^t, \lambda_i^t) - \mathcal{L}_i(\mathbf{x}_i^{t+1}, \mathbf{z}^{t+1}, \lambda_i^{t+1})\right]$$

$$+ \frac{\lambda_{\max}}{N}\sum_{i=1}^N \|\mathbf{x}_i^{t+1} - \mathbf{z}^{t+1}\|^2 + \frac{1}{N}\sum_{i=1}^N \frac{M^2\epsilon}{2L_i} + \alpha M^2 G^2. \tag{75}$$

Using (75) into (74), we get

$$\mathbb{E}[\|\nabla f(\mathbf{z}^t)\|^2] \leq \frac{M}{N}\sum_{i=1}^N \mathbb{E}[\|\mathbf{z}^t - \mathbf{x}_i^t\|^2] + \frac{4M^2}{N}\sum_{i=1}^N \mathbb{E}\left[\mathcal{L}_i(\mathbf{x}_i^t, \mathbf{z}^t, \lambda_i^t) - \mathcal{L}_i(\mathbf{x}_i^{t+1}, \mathbf{z}^{t+1}, \lambda_i^{t+1})\right]$$

$$+ \frac{2\lambda_{\max}}{N}\sum_{i=1}^N \|\mathbf{x}_i^{t+1} - \mathbf{z}^{t+1}\|^2 + \frac{1}{N}\sum_{i=1}^N \frac{M^2\epsilon}{2L_i} + 2\alpha M^2 G^2. \tag{76}$$

Take the summation over $t = 0$ to $T - 1$ to obtain

$$\sum_{t=0}^{T-1}\mathbb{E}[\|\nabla f(\mathbf{z}^t)\|^2] \leq \left(\frac{M + 2\lambda_{\max}}{N}\right)\sum_{t=0}^{T-1}\sum_{i=1}^N \mathbb{E}[\|\mathbf{z}^t - \mathbf{x}_i^t\|^2] + \frac{4M^2}{N}\sum_{i=1}^N \mathbb{E}\left[\mathcal{L}_i(\mathbf{x}_i^0, \mathbf{z}^0, \lambda_i^0) - \mathcal{L}_i(\mathbf{x}_i^T, \mathbf{z}^T, \lambda_i^T)\right]$$

$$+ \frac{2\lambda_{\max}}{N}\sum_{i=1}^N \|\mathbf{x}_i^T - \mathbf{z}^T\|^2 + \frac{1}{N}\sum_{i=1}^N \frac{M^2\epsilon T}{2L_i} + 2\alpha M^2 G^2 T. \tag{77}$$

Let us recall the expression in (77) as follows

$$\sum_{t=0}^{T-1} \mathbb{E}[\|\nabla f(\mathbf{z}^t)\|^2] \leq \frac{M}{N} \sum_{i=1}^{N} \mathbb{E}[\|\mathbf{z}^0 - \mathbf{x}_i^0\|^2] + \frac{4M^2}{N} \sum_{i=1}^{N} \mathbb{E}\left[\mathcal{L}_i(\mathbf{x}_i^0, \mathbf{z}^0, \lambda_i^0) - \mathcal{L}_i(\mathbf{x}_i^T, \mathbf{z}^T, \lambda_i^T)\right]$$
$$+ \left(\frac{M + 2\lambda_{\max}}{N}\right) \sum_{t=0}^{T-1} \sum_{i=1}^{N} \mathbb{E}[\|\mathbf{z}^{t+1} - \mathbf{x}_i^{t+1}\|^2] + \frac{1}{N} \sum_{i=1}^{N} \frac{M^2 \epsilon T}{2L_i} + 2\alpha M^2 G^2 T. \qquad (78)$$

We assume that the initialization is such that $\mathbf{x}_i^0 = \mathbf{z}^0$ for all $i$. This implies that

$$\sum_{t=0}^{T-1} \mathbb{E}[\|\nabla f(\mathbf{z}^t)\|^2] \leq \frac{4M^2}{N} \sum_{i=1}^{N} \mathbb{E}\left[\mathcal{L}_i(\mathbf{x}_i^0, \mathbf{z}^0, \lambda_i^0) - \mathcal{L}_i(\mathbf{x}_i^T, \mathbf{z}^T, \lambda_i^T)\right] \qquad (79)$$
$$+ \left(\frac{M + 2\lambda_{\max}}{N}\right) \sum_{t=0}^{T-1} \sum_{i=1}^{N} \mathbb{E}[\|\mathbf{z}^{t+1} - \mathbf{x}_i^{t+1}\|^2] + \frac{1}{N} \sum_{i=1}^{N} \frac{M^2 \epsilon T}{2L_i} + 2\alpha M^2 G^2 T.$$

Dividing both sides by $T$ and then utilizing the upper bound from the statement of Lemma A.2 (cf. (47)) into the right-hand side of (79), we get

$$\frac{1}{T} \sum_{t=0}^{T-1} \mathbb{E}[\|\nabla f(\mathbf{z}^t)\|^2] \leq \frac{4M^2}{NT} \sum_{i=1}^{N} \mathbb{E}\left[\mathcal{L}_i(\mathbf{x}_i^0, \mathbf{z}^0, \lambda_i^0) - \mathcal{L}_i(\mathbf{x}_i^T, \mathbf{z}^T, \lambda_i^T)\right]$$
$$+ \frac{(M + 2\lambda_{\max})\epsilon S(N-1)}{C_0^2 N^2} \sum_{i=1}^{N} \frac{1}{(\beta L_i)^2} + \frac{\delta^2 S(N-1)}{C_0^2 N^2} \sum_{i=1}^{N} \frac{1}{(\beta L_i)^4}$$
$$+ \frac{1}{N} \sum_{i=1}^{N} \frac{M^2 \epsilon}{2L_i} + 2\alpha M^2 G^2. \qquad (80)$$

From the initial conditions, we have $\mathcal{L}_i(\mathbf{x}_i^0, \mathbf{z}^0, \lambda_i^0) = f_i(\mathbf{z}^0)$. Now we need a lower bound on $\mathcal{L}_i(\mathbf{x}_i^T, \mathbf{z}^T, \lambda_i^T)$, we can write from the definition of Lagrangian the following

$$\mathcal{L}_i(\mathbf{x}_i^T, \mathbf{z}^T, \lambda_i^T) = f_i(\mathbf{x}_i^T) + \lambda_i^T \left(\|\mathbf{x}_i^T - \mathbf{z}^T\|^2 - \gamma_i\right). \qquad (81)$$

After rearranging the terms, we get

$$f_i(\mathbf{x}_i^T) = \mathcal{L}_i(\mathbf{x}_i^T, \mathbf{z}^T, \lambda_i^T) - \lambda_i^T \|\mathbf{x}_i^T - \mathbf{z}^T\|^2 + \gamma_i \lambda_i^T. \qquad (82)$$

From the smoothness assumption (cf. Assumption 4.2), we note that

$$f_i(\mathbf{z}^T) \leq f_i(\mathbf{x}_i^T) + \langle \nabla f_i(\mathbf{x}_i^T), \mathbf{z}^T - \mathbf{x}_i^T \rangle + \frac{L_i}{2} \|\mathbf{z}^T - \mathbf{x}_i^T\|^2. \qquad (83)$$

Substitute the value in (82) into (83), we get

$$f_i(\mathbf{z}^T) \leq \mathcal{L}_i(\mathbf{x}_i^T, \mathbf{z}^T, \lambda_i^T) - \lambda_i^T \|\mathbf{x}_i^T - \mathbf{z}^T\|^2 + \gamma_i \lambda_i^T + \langle \nabla f_i(\mathbf{x}_i^T), \mathbf{z}^T - \mathbf{x}_i^T \rangle + \frac{L_i}{2} \|\mathbf{z}^T - \mathbf{x}_i^T\|^2$$
$$\leq \mathcal{L}_i(\mathbf{x}_i^T, \mathbf{z}^T, \lambda_i^T) - \lambda_i^T \|\mathbf{x}_i^T - \mathbf{z}^T\|^2 + \gamma_i \lambda_i^T + \frac{G^2}{2L_i} + \frac{L_i}{2} \|\mathbf{z}^T - \mathbf{x}_i^T\|^2 + \frac{L_i}{2} \|\mathbf{z}^T - \mathbf{x}_i^T\|^2$$
$$\leq \mathcal{L}_i(\mathbf{x}_i^T, \mathbf{z}^T, \lambda_i^T) - (\lambda_i^T - L_i)\|\mathbf{x}_i^T - \mathbf{z}^T\|^2 + \gamma_i \lambda_i^T + \frac{G^2}{2L_i}, \qquad (84)$$

where we utilize $\|\nabla f_i(\mathbf{x}_i^T)\| \leq G$ in the second inequality. Since, $\lambda_i^T \geq L_i$, dropping the negative terms from the right hand side of (68), we get

$$f_i(\mathbf{z}^T) \leq \mathcal{L}_i(\mathbf{x}_i^T, \mathbf{z}^T, \lambda_i^T) + \gamma_i \lambda_i^T + \frac{G^2}{2L_i}. \qquad (85)$$

After rearranging the terms, we get

$$\mathcal{L}_i(\mathbf{x}_i^T, \mathbf{z}^T, \lambda_i^T) \geq f_i(\mathbf{z}^T) - \gamma_i \lambda_i^T - \frac{G^2}{2L_i}. \tag{86}$$

Multiply both sides by $-1$, we get

$$-\mathcal{L}_i(\mathbf{x}_i^T, \mathbf{z}^T, \lambda_i^T) \leq -f_i(\mathbf{z}^T) + \gamma_i \lambda_i^T + \frac{G^2}{2L_i}. \tag{87}$$

Add $\mathcal{L}_i(\mathbf{x}_i^0, \mathbf{z}^0, \lambda_i^0) = f_i(\mathbf{z}^0)$ to both sides, we get

$$\mathcal{L}_i(\mathbf{x}_i^0, \mathbf{z}^0, \lambda_i^0) - \mathcal{L}_i(\mathbf{x}_i^T, \mathbf{z}^T, \lambda_i^T) \leq f_i(\mathbf{z}^0) - f_i(\mathbf{z}^T) + \gamma_i \lambda_i^T + \frac{G^2}{2L_i}. \tag{88}$$

Take average across agents to obtain

$$\frac{1}{N}\sum_{i=1}^{N}[\mathcal{L}_i(\mathbf{x}_i^0, \mathbf{z}^0, \lambda_i^0) - \mathcal{L}_i(\mathbf{x}_i^T, \mathbf{z}^T, \lambda_i^T)] \leq \frac{1}{N}\sum_{i=1}^{N}[f_i(\mathbf{z}^0) - f_i(\mathbf{z}^T)] + \frac{1}{N}\sum_{i=1}^{N}\gamma_i \lambda_i^T + \frac{1}{N}\sum_{i=1}^{N}\frac{G^2}{2L_i}. \tag{89}$$

Next, for the optimal global model $\mathbf{z}^\star$, it would hold that $-\frac{1}{N}\sum_{i=1}^{N} f_i(\mathbf{z}^T) \leq -\frac{1}{N}\sum_{i=1}^{N} f_i(\mathbf{z}^\star)$, we can write

$$\frac{1}{N}\sum_{i=1}^{N}[\mathcal{L}_i(\mathbf{x}_i^0, \mathbf{z}^0, \lambda_i^0) - \mathcal{L}_i(\mathbf{x}_i^T, \mathbf{z}^T, \lambda_i^T)] \leq \frac{1}{N}\sum_{i=1}^{N}[f_i(\mathbf{z}^0) - f_i(\mathbf{z}^\star)] + \frac{\lambda_{\max}}{N}\sum_{i=1}^{N}\gamma_i + \frac{1}{N}\sum_{i=1}^{N}\frac{G^2}{2L_i}. \tag{90}$$

In the above inequality, let us define the constant

$$B_0 := \frac{1}{N}\sum_{i=1}^{N}[f_i(\mathbf{z}^0) - f_i(\mathbf{z}^\star)] + \frac{1}{N}\sum_{i=1}^{N}\frac{G^2}{2L_i}, \tag{91}$$

we can write

$$\frac{1}{N}\sum_{i=1}^{N}[\mathcal{L}_i(\mathbf{x}_i^0, \mathbf{z}^0, \lambda_i^0) - \mathcal{L}_i(\mathbf{x}_i^T, \mathbf{z}^T, \lambda_i^T)] \leq B_0 + \frac{\lambda_{\max}}{N}\sum_{i=1}^{N}\gamma_i. \tag{92}$$

Using the similar argument in (40)-(46), after taking expectation with respect to the randomness due to device sampling, and then taking total expectation, we can write

$$\frac{1}{N}\sum_{i=1}^{N}\mathbb{E}[\mathcal{L}_i(\mathbf{x}_i^0, \mathbf{z}^0, \lambda_i^0) - \mathcal{L}_i(\mathbf{x}_i^T, \mathbf{z}^T, \lambda_i^T)] \leq B_0 + \frac{\lambda_{\max}}{N}\sum_{i=1}^{N}\gamma_i. \tag{93}$$

Utilize the upper bound of (92) into the right hand side of (80) to obtain

$$\frac{1}{T}\sum_{t=0}^{T-1}\mathbb{E}[\|\nabla f(\mathbf{z}^t)\|^2] \leq \frac{4M^2 B_0}{T} + \frac{4M^2\lambda_{\max}}{T}\frac{1}{N}\sum_{i=1}^{N}\gamma_i + \frac{(M + 2\lambda_{\max})\epsilon(S(N-1))}{C_0^2 N^2}\sum_{i=1}^{N}\frac{1}{(\beta L_i)^2}$$
$$+ \frac{\delta^2 S(N-1)}{C_0^2 N^2}\sum_{i=1}^{N}\frac{1}{(\beta L_i)^4} + \frac{1}{N}\sum_{i=1}^{N}\frac{M^2\epsilon}{2L_i} + 2\alpha M^2 G^2. \tag{94}$$

Next, in the order notation, we could write

$$\frac{1}{T}\sum_{t=0}^{T-1}\mathbb{E}[\|\nabla f(\mathbf{z}^t)\|^2] \leq \mathcal{O}\left(\frac{M^2 B_0}{T}\right) + \mathcal{O}\left(\frac{M^2\lambda_{\max}}{TN}\sum_{i=1}^{N}\gamma_i\right) + \mathcal{O}\left(\frac{(M + 2\lambda_{\max})\epsilon S}{C_0^2 N}\sum_{i=1}^{N}\frac{1}{(\beta L_i)^2}\right)$$
$$+ \mathcal{O}\left(\frac{\delta^2 S}{C_0^2 N}\sum_{i=1}^{N}\frac{1}{(\beta L_i)^4}\right) + \mathcal{O}\left(\frac{M^2\epsilon}{L}\right) + \mathcal{O}\left(\frac{1}{N}\sum_{i=1}^{N}\frac{M^2\epsilon}{2L_i}\right). \tag{95}$$

Hence proved. $\qquad\square$

# C   Proof of Corollary 4.7

*Proof.* Here we prove the effect of the introduced parameter $\gamma_i$ on the local/personalized performance of the proposed algorithm. The purpose of this analysis is to develop an intuitive explanation for the relationship between $\gamma_i$ and the localized performance. We emphasize that the analysis here relies on a relaxed version (cf. (99)) of the dual update mentioned in (11). For the localized performance, we analyze how $\nabla f(\mathbf{x}_i^{t+1})$ behaves. From (26), it holds that

$$\nabla f_i(\mathbf{x}_i^{t+1}) = \mathbf{e}_i^{t+1} - \left(2\lambda_i^t\right)\left(\mathbf{x}_i^{t+1} - \mathbf{z}^t\right). \tag{96}$$

Taking norm on both sides and utilizing the definition of $\epsilon_i$-approximate local solution, we can write

$$\|\nabla f_i(\mathbf{x}_i^{t+1})\|^2 \le 2\epsilon_i + 8|\lambda_i^t|^2 \cdot \|\mathbf{x}_i^{t+1} - \mathbf{z}^t\|^2. \tag{97}$$

Next, utilizing Assumption 4.1, we can conclude that $\|\mathbf{x}_i^{t+1} - \mathbf{z}^t\| \le 2R$, which eventually implies that

$$\|\nabla f_i(\mathbf{x}_i^{t+1})\|^2 \le 2\epsilon_i + 32R^2|\lambda_i^t|^2, \tag{98}$$

where we obtain a loose upper bound on the term $\|\nabla f_i(\mathbf{x}_i^{t+1})\|^2$ to understand it's behaviour intuitively. Since the whole idea of dual variable $\lambda_i^t$ is to act as a penalty parameter if the constraint in (1) is not satisfied, we consider a relaxed intuitive version of the dual update in (11) where we remove the projection operator and consider the update

$$\lambda_i^t = \left[\lambda_i^{t-1} + \alpha \mathbb{I}_{\{\|\mathbf{x}_i^t - \mathbf{z}^{t-1}\|^2 \le \gamma_i)\}}\right]_+, \tag{99}$$

where

$$\mathbb{I}_{\{\|\mathbf{x}_i^t - \mathbf{z}^{t-1}\|^2 > \gamma_i\}} = \begin{cases} 1, & \text{if } \|\mathbf{x}_i^t - \mathbf{z}^{t-1}\|^2 > \gamma_i \\ -1, & \text{if } \|\mathbf{x}_i^t - \mathbf{z}^{t-1}\|^2 \le \gamma_i. \end{cases} \tag{100}$$

Hence, we increase the penalty by $\alpha$ if the constraint is not satisfied and decrease the penalty by $\alpha$ if the constraint is satisfied. After unrolling the recursion in (99), we can write

$$\lambda_i^t \le \alpha \left[\sum_{k=0}^t \mathbb{I}_{\{\|\mathbf{x}_k^t - \mathbf{z}^{k-1}\|^2 \le \gamma_i)\}}\right]_+. \tag{101}$$

Substitute (101) into the right hand side of (98) to obtain

$$\|\nabla f_i(\mathbf{x}_i^{t+1})\|^2 \le \mathcal{O}\left(\epsilon_i\right) + \alpha^2 \mathcal{O}\left(\left[\sum_{k=0}^t \mathbb{I}_{\{\|\mathbf{x}_i^k - \mathbf{z}^{k-1}\|^2 \le \gamma_i)\}}\right]_+\right)^2. \tag{102}$$

The above expression holds for any $t$. Hence, write it for $t$, take summation over $t = 1$ to $T$ and then divide by $T$ to obtain

$$\frac{1}{T}\sum_{t=1}^T \|\nabla f_i(\mathbf{x}_i^t)\|^2 \le \mathcal{O}\left(\epsilon_i\right) + \alpha^2 \frac{1}{T}\sum_{t=1}^T \mathcal{O}\left(\left[\sum_{k=0}^t \mathbb{I}_{\{\|\mathbf{x}_i^k - \mathbf{z}^{k-1}\|^2 \le \gamma_i)\}}\right]_+\right)^2 \tag{103}$$

$$\le \mathcal{O}\left(\epsilon_i\right) + \alpha^2 \mathcal{O}\left(\left[\sum_{k=0}^T \mathbb{I}_{\{\|\mathbf{x}_i^k - \mathbf{z}^{k-1}\|^2 \le \gamma_i)\}}\right]_+\right)^2. \tag{104}$$

Hence proved □

# D  Additional Details of the Experiments

## D.1  Experiment Setup

**Datasets.** We perform experiments on both synthetic and real datasets. The devices are heterogeneous due to non-independent and identically distributed (non-iid) data. The degree of correlation is manifested through the heterogeneous class distribution across devices and the variability in the size of the local dataset at each device.

- The **synthetic dataset** is for a 10-class classification task, and is adapted from (Li et al., 2020) with parameters $\alpha$ and $\beta$ controlling model and data variations across devices. Specifically, each device's local model is a multinomial logistic regression of the form $y_i = \mathrm{argmax}(\mathrm{softmax}(\mathbf{W_i}\mathbf{x_i} + \mathbf{b_i}))$ with $\mathbf{W_i} \in \mathbb{R}^{10\times 60}$ and $\mathbf{b_i} \in \mathbb{R}^{10}$ as in (Li et al., 2020). The local data $(\mathbf{x_i}, y_i)$ of each device is generated according to (Li et al., 2020). We sample $\mathbf{W_i} \sim \mathcal{N}(u_i, 1)$ and $\mathbf{b_i} \sim \mathcal{N}(u_i, 1)$ where $u_i \sim \mathcal{N}(0, \alpha)$. We sample $\mathbf{x_i}$ from $\mathcal{N}(\mathbf{v_i}, \mathbf{\Sigma})$, where $\mathbf{\Sigma}$ is a diagonal matrix with entries $\mathbf{\Sigma}_{j,j} = j^{-1.2}$, and each entry of $\mathbf{v_i}$ is from $\mathcal{N}(B_k, 1)$, $B_k \sim \mathcal{N}(0, \beta)$. We set $\alpha = \beta = 0.5$, and partition the data among 30 devices according to a power law.

- For **real datasets**, we use MNIST and CIFAR-10 for image classification and the Shakespeare dataset for the next character prediction task. MNIST (LeCun et al., 1998) and CIFAR-10 (Krizhevsky et al., 2009) datasets consist of handwritten digits and color images from 10 different classes. We follow the strategy from (Stripelis & Ambite, 2020) to distribute the data to 20 devices following a power law with a preset exponent (Data size range and distribution for each exponent are shown in Table 5 and Figure 5). To preserve the power law relationship, we allow some devices to have more classes than others, as in (Stripelis & Ambite, 2020). Table 4 summarizes the number of classes each device has for different power law exponents. We denote $C$ to represent the most common number of classes owned by devices. For example, CIFAR-10 with $C = 2$ and exponent 1.2 is distributed as $\{1 \times 5, 2 \times 4, 17 \times 2\}$, which means 1 device has 5 classes, 2 devices have 4 classes, and the rest 17 devices have 2 classes. Besides computer vision tasks, we also perform experiments on natural language processing tasks such as next-character prediction on the Shakespeare dataset from (McMahan et al., 2017). The dataset is composed of character sequences from different speaking roles. We assign each speaking role to a device, and we subsample 138 device.

**Architectures.** For the MNIST dataset, we use a neural network consisting of 2 fully-connected layers with 128 hidden dimensions. For the CIFAR-10 dataset, we use a 2-layer convolutional neural network with 5 as kernel size. The first and second layers have 6 and 16 filters, respectively (Li et al., 2021a). After each convolution operation, we apply $2 \times 2$ max pooling and Relu activation. For the Shakespeare dataset, we use a 2-layer LSTM with 256 as the dimension of the hidden state. The input to this model is a character sequence of length 80, and we embed it into a vector of size 8 before passing it into the LSTM (Li et al., 2019).

**Baselines.** We use the term global accuracy when evaluating the performance of the global model using the entire test dataset. We use the term local accuracy when evaluating the performance of each device's local model using its own test data and take the average across all devices. To evaluate the global accuracy of `FedBC` in terms of the prediction accuracy of test data across all the devices, we compare it with 6 other federated learning algorithms, i.e., FedAvg (McMahan et al., 2017), q-FedAvg (Li et al., 2019), FedProx (Li et al., 2020), Scaffold (Karimireddy et al., 2020), Per-FedAvg (Fallah et al., 2020) and pFedMe (T Dinh et al., 2020). For Per-FedAvg and pFedMe, we us the model at the server to evaluate the global performance across all the devices. Besides the baseline algorithms mentioned above, we compare the performance of local models of `FedBC` with personalization algorithms such as Per-FedAvg (Fallah et al., 2020) and pFedMe (T Dinh et al., 2020). To further enhance the performance of local models, we fine-tune the local models of `FedBC` by performing 1-step gradient descent using a batch of test data and name this approach as `FedBC`-FineTune. In the end, as an experimental study, we also incorporate MAML-type training into `FedBC`, and propose algorithm Per-`FedBC` (see appendix D.4). Similar to `FedBC`-FineTune, we also perform the 1-step fine-tuning step using test data for Per-`FedBC` or Per-FedAvg following (Fallah et al., 2020).

## D.2 Hyperparameters Tuning

For FedAvg and Scaffold, we search the learning rate $(lr)$ in the range $[0.001, 0.01, 0.1, 0.5, 1.0]$. For q-FedAvg, FedProx, and `FedBC`, besides tuning $lr$ in this range, we also tune other algorithmic-specific parameters. For Fedprox, we tune parameter $\mu$ in the range $[0.0001, 0.001, 0.01, 0.1, 1.0]$ (see (4) for the definition of $\mu$). For q-FedAvg, we tune $q$ in the range $[0.001, 0.01, 0.1, 1.0, 2.0, 5.0]$ (see (Li et al., 2019) for the definition $q$). For `FedBC`, we tune the learning rate for the Lagrangian multiplier $\lambda$ $(lr_\lambda)$ when updating it through gradient ascent in the range $[10^{-7}, 10^{-6}, 10^{-5}, 0.0001, 0.001, 0.01]$. As mentioned in D.1, we also perform a gradient-descent type update for the constant $\gamma_i$. We set its learning rate equal to $lr_\lambda$ to limit the search cost, and it works well in practice. For Per-FedAvg, we choose the inner-level learning rate $(\alpha)$ and outer-level learning rate $(\beta)$ to be 0.01 and 0.001 as in (Fallah et al., 2020). For Per-`FedBC` (see Algorithm 3), we set $\alpha$ and $\beta$ to be the same as Per-FedAvg with $J = 1$. For pFedMe, we tune parameter $\mu$ in the range $[0.01, 0.1, 1.0, 10.0]$ (see (5) for the definition of $\mu$). We subsample 10 devices at each round of communication for all algorithms and perform 5 local training epochs unless otherwise stated (we enote the number of local training epochs as $E$). Once the optimal hyperparameter is found, we perform 5 parallel runs to compute the standard deviation for each algorithm. We use an 80%/20% train/test split for all datasets.

## D.3 Data Partitioning

Table 4 summarizes the class partitioning for MNIST and CIFAR-10 datasets. In this table, $C$ represents the most common number of classes owned by devices. Devices in the head of the power law data size distribution are given more classes than others as in (Stripelis & Ambite, 2020). For example, $C = 1$ with power law exponent 1.1 for MNIST has the class partitioning $\{2 \times 3, 5 \times 2, 13 \times 1\}$, which means 2 devices have 3 classes, 5 devices have 2 classes, and 13 devices have 1 classes. Table 5 summarizes the range of data sizes for different power law exponents. For example, for CIFAR-10 with a power law exponent 1.1, the number of data a device has is between 1047 and 6433. Figure 5 and Figure 6 further show the number of data each device has for real and synthetic datasets, respectively.

| Classes | Dataset | Power Law Exponent | | | | |
|---|---|---|---|---|---|---|
| | | 1.1 | 1.2 | 1.3 | 1.4 | 1.5 |
| $c = 1$ | MNIST | $\{2 \times 3, 5 \times 2, 13 \times 1\}$ | $\{1 \times 4, 1 \times 3, 4 \times 2, 14 \times 1\}$ | $\{1 \times 4, 1 \times 3, 4 \times 2, 14 \times 1\}$ | $\{1 \times 6, 1 \times 3, 2 \times 2, 16 \times 1\}$ | $\{1 \times 6, 1 \times 3, 2 \times 2, 16 \times 1\}$ |
| | CIFAR-10 | $\{2 \times 3, 5 \times 2, 13 \times 1\}$ | $\{1 \times 6, 1 \times 3, 4 \times 2, 14 \times 1\}$ | $\{1 \times 5, 1 \times 3, 2 \times 2, 16 \times 1\}$ | $\{1 \times 6, 1 \times 3, 2 \times 2, 16 \times 1\}$ | $\{1 \times 6, 1 \times 3, 2 \times 2, 16 \times 1\}$ |
| $c = 2$ | MNIST | $\{1 \times 5, 1 \times 4, 3 \times 3, 15 \times 2\}$ | $\{1 \times 6, 1 \times 4, 1 \times 3, 17 \times 2\}$ | $\{1 \times 6, 1 \times 4, 1 \times 3, 17 \times 2\}$ | $\{1 \times 6, 1 \times 4, 1 \times 3, 17 \times 2\}$ | $\{1 \times 6, 1 \times 4, 1 \times 3, 17 \times 2\}$ |
| | CIFAR-10 | $\{1 \times 5, 1 \times 4, 3 \times 3, 15 \times 2\}$ | $\{1 \times 5, 2 \times 4, 17 \times 2\}$ | $\{1 \times 6, 1 \times 4, 1 \times 3, 17 \times 2\}$ | $\{1 \times 6, 1 \times 4, 1 \times 3, 17 \times 2\}$ | $\{1 \times 6, 1 \times 4, 1 \times 3, 17 \times 2\}$ |
| $c = 3$ | MNIST | $\{1 \times 7, 2 \times 4, 17 \times 3\}$ | $\{1 \times 7, 2 \times 4, 17 \times 3\}$ | $\{1 \times 8, 2 \times 4, 17 \times 3\}$ | $\{1 \times 8, 2 \times 4, 17 \times 3\}$ | $\{1 \times 8, 2 \times 4, 17 \times 3\}$ |
| | CIFAR-10 | $\{2 \times 5, 1 \times 4, 17 \times 3\}$ | $\{1 \times 7, 2 \times 4, 17 \times 3\}$ | $\{1 \times 8, 2 \times 4, 17 \times 3\}$ | $\{1 \times 8, 2 \times 4, 17 \times 3\}$ | $\{1 \times 9, 1 \times 4, 18 \times 3\}$ |

Table 4: Class partitioning of MNIST and CIFAR-10 for different power law exponents. Total number of devices is 20.

| Dataset | Power Law Exponent | | | | |
|---|---|---|---|---|---|
| | 1.1 | 1.2 | 1.3 | 1.4 | 1.5 |
| MNIST | [1222 7486] | [374 12057] | [110 16444] | [32 21197] | [10 23410] |
| CIFAR-10 | [1047 6433] | [320 10398] | [94 14032] | [28 17814] | [9 20067] |

Table 5: Data size ranges of MNIST and CIFAR-10 for different power law exponents. The total number of devices is 20.

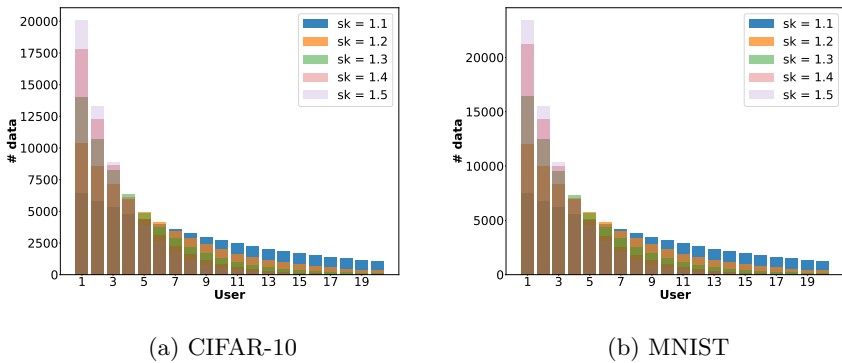

(a) CIFAR-10          (b) MNIST

Figure 5: (a) Number of CIFAR-10 and (b) MNIST data each device has for different power law exponents as indicated by the legend. The total number of devices is 20.

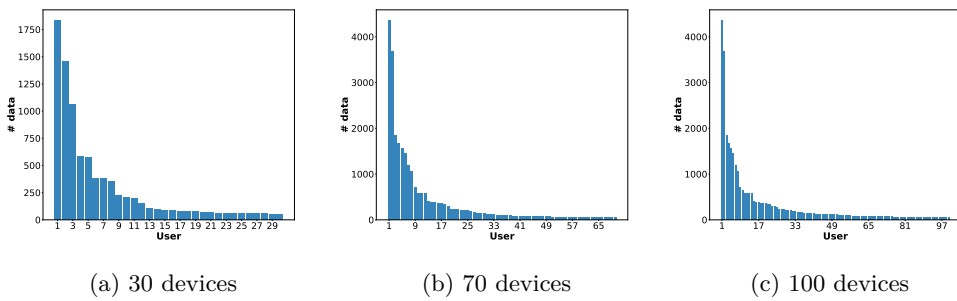

(a) 30 devices      (b) 70 devices      (c) 100 devices

Figure 6: Distribution of synthetic data across devices. (a) 30 devices, (b) 70 devices, and (c) 100 devices.

## D.4 Per-`FedBC` Algorithm

The idea of MAML is to find a good global initializer that can be quickly adapted to new tasks (Fallah et al., 2020), i.e., solve the objective of (6). Its derivative involves second-order terms, which creates computational burden. A first-order approximation, i.e. FO-MAML, ignores them to avoid this problem. More generally, a MAML problem can be viewed as a bilevel optmization problem in which the upper-level objective is to find and update this initializer, and the lower-level objective is to fine-tune the initializer based on specific task information (Fan et al., 2021). We utilize this idea to propose an advanced version of `FedBC` called Per-`FedBC` for experimental purposes.

Due to the constraint in (6), we treat the inner objective of the form $\mathcal{L}_i(\mathbf{z}, \mathbf{x}_i, \lambda_i) := \left[ f_i(\mathbf{x}_i) + \lambda_i \left( \|\mathbf{x}_i - \mathbf{z}\|^2 - \gamma_i \right) \right]$. Hence, when we update each device's local model based on this objective, it has a term associated with the global model $\mathbf{z}^t$. We also ignore any second-order derivatives to reduce computation cost, hence different from Per-FedAvg. First, the local updates involve Lagrange multipliers due to constraint penalization. Second, we also perform the dual updates as in `FedBC`. Third, we follow the aggregation strategy based on Lagrange multipliers as in `FedBC` to update the global model. We summarize the steps in Algorithm 3 and report the performance of Per-`FedBC` in Table 3.

## E Additional Experiments on Synthetic Dataset

### E.1 Training loss for different Epochs $E$ and Lagrangian visualization

Figure 7 shows the training loss for different $E$s. We observe that `FedBC` achieves the lowest train loss for $E = 1, 10, 25$ and $50$ compared to other algorithms. These observations are consistent with Figure 3a. We

---

**Algorithm 3** Personalized Federated Beyond Consensus (Per-`FedBC`)

---

1: **Input**: $T$, $K$, $J$, $\alpha$, $\alpha_\lambda$, and $\beta$.
2: **Initialize**: $\mathbf{z}^0$, $\gamma_i^0$, and $\lambda_i^0$ for all $i$.
3: **for** $t = 0$ to $T - 1$ **do**
4:    Sub-sample a set of devices of size M; for each device i, perform the following updates
5:    $\mathbf{w}_i^0 = \mathbf{z}^t$
6:    **for** $k = 0$ to $K - 1$ **do**
7:       $\mathbf{w}_{i,0}^k = \mathbf{w}_i^k$
8:       **for** $j = 0$ to $J - 1$ **do**
9:

$$\mathbf{w}_{i,j+1}^k = \mathbf{w}_{i,j}^k - \alpha \left( \nabla_{\mathbf{w}} f_i(\mathbf{w}_{i,j}^k) + (2\lambda_i^t) \left( \mathbf{w}_{i,j}^k - \mathbf{z}^t \right) \right)$$

10:       **end for**
11:       $\mathbf{w}_i^{k+1} = \mathbf{w}_i^k - \beta \nabla_{\mathbf{w}} f_i(\mathbf{w}_{i,J}^k)$
12:    **end for**
13:    Primal update: $\mathbf{x}_i^{t+1} = \mathbf{w}_i^K$
14:    Dual update: $\lambda_i^{t+1} = \mathcal{P}_\Lambda \left[ \lambda_i^t + \alpha_\lambda (\|\mathbf{x}_i^{t+1} - \mathbf{z}^t\|^2 - \gamma_i) \right]$
15:    **Server** update

$$\mathbf{z}^{t+1} = \frac{1}{\sum_{j \in \mathcal{S}_t} \lambda_j^{t+1}} \sum_{j \in \mathcal{S}_t} (\lambda_j^{t+1} \mathbf{x}_j^{t+1})$$

16: **end for**
17: **Output**: $\mathbf{z}^T$

---

also observe some divergence in the training loss of the Scaffold for high $E = 25$ and 50. Figure 8 shows the coefficient of min and max user/device when computing the global model for $E = 1, 10, 25$ and 50. We observe that the coefficient based on the Lagrangian multiplier is consistently less biased towards the min user than its data-size-based counterpart for all $E$s, which provides additional evidence besides Figure 3b to show that the devices of `FedBC` participate fairly in updating the global model. This improves the performance disparity of the global model obtained by `FedBC`.

**Lagrangian visualization.** Since we are introducing Lagrangian variables via `FedBC`, we further track the Lagrangian multipliers ($\lambda$) at different rounds of communication and plot their magnitudes for all devices in 7e. We observe that devices of small data sizes can have $\lambda$s that are very similar in magnitudes to devices of large data sizes at different rounds of communication. This shows that the fair aggregation of local models for `FedBC` occurs throughout the training process and not only in the end as shown in Figure 3c. The results in Table 1 is based on 30 total number of devices. Here, we provide additional results for the different total numbers of devices as shown in Figure 7f. We observe that `FedBC` outperforms other algorithms when the total number of devices is 30 or 70. `FedBC` and Scaffold have similar performance and outperform others when this becomes 100.

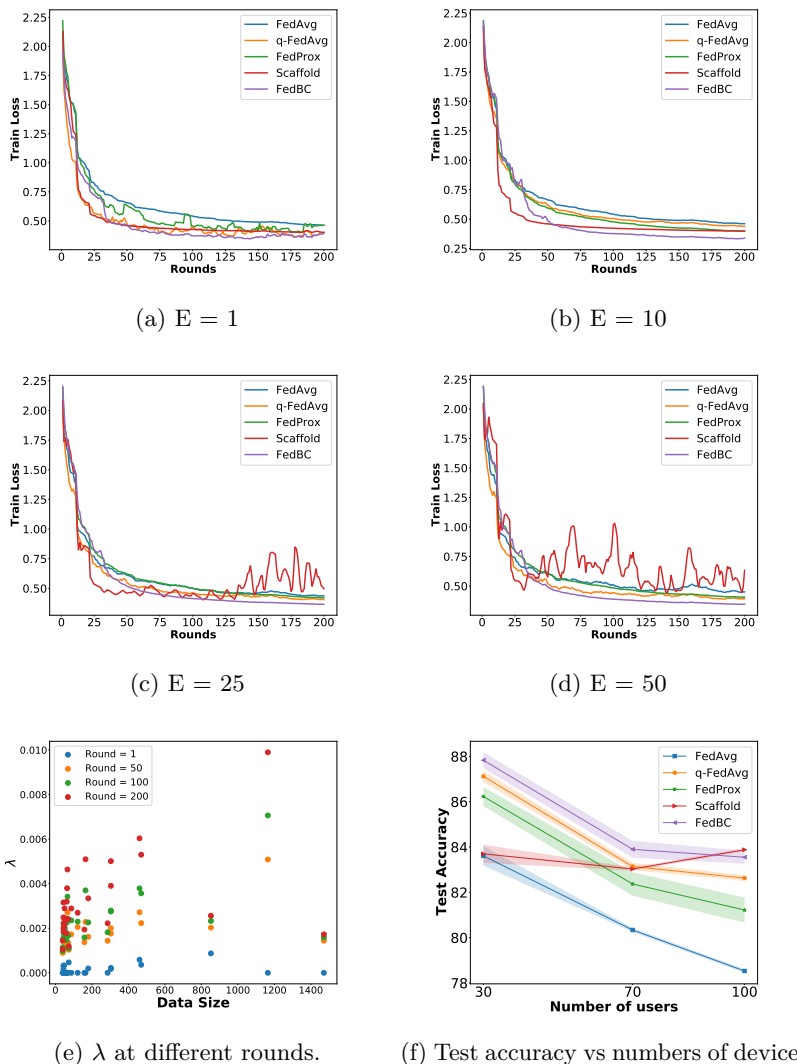

(a) E = 1

(b) E = 10

(c) E = 25

(d) E = 50

(e) $\lambda$ at different rounds.

(f) Test accuracy vs numbers of devices.

Figure 7: In **(a)**-**(d)**, we show training loss of FedAvg, q-FedAvg, FedProx, and Scaffold for (a) $E = 1$, (b) $E = 10$, (c) $E = 25$, and (d) $E = 50$.
**(e)** We show $\lambda$ at round 1, 50, 100, and 200 of communication for devices of different data size. We observe for the majority of devices, changes in $\lambda$ happen in the first 50 rounds. **(f)** We plot test accuracy against the total number of devices. The shaded region shows a standard deviation. We observe that `FedBC` outperforms all other algorithms when the total number of devices is 30 or 70. For 100 devices, `FedBC` has similar performance as Scaffold.

### E.1.1 Evidence of Reducing Performance Disparity Induced by `FedBC`

Figure 3 and Figure 8 together show the global model of `FedBC` incorporates feedback from local models in an unbiased way. In Figure 8, we plot the coefficient which decides the contribution of min and max user towards the global model and plots it against the number of communication rounds. We plot it for different values of $E$. We note from Figure 8 that the contribution coefficients are (red and green) becoming similar as the communication rounds increase, ensuring the unbiased nature of the `FedBC` algorithm. Because of this, it can perform well on both the min and max device's local data, as shown in Figure 4. We can see the difference between the final values of data size-based contribution coefficients (orange and blue), which are quite different, resulting in biased behavior.

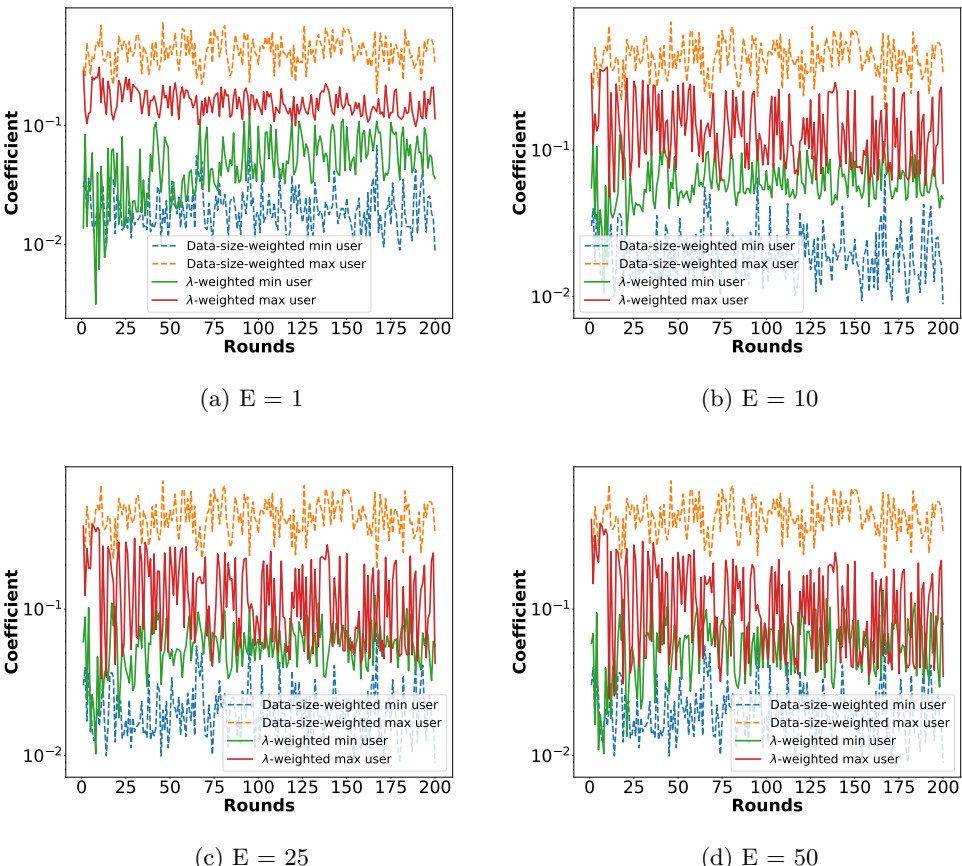

Figure 8: Coefficient of min and max user based on either Lagrangian multipliers or data sizes when computing the global model for $E = 1$ (a), 10 (b), 25 (c), and 50 (d). Min and max users are defined in Figure 3 caption. They differ from one communication round to another due to random sampling of 10 devices.

**Test accuracy of min and max user/device.** To further solidify our claim, we plot the test accuracy of min and max user/device in Figure 9-12 for $E = 1, 10, 25$ and 50, respectively. We observe that `FedBC` is the best in eliminating the difference in test accuracy of min and max user's local data for all $E$s. We also notice this difference is most apparent when $E = 50$ for all algorithms as shown in Figure 12. As $E$ increases, each device's local model would differ more from the global model. This creates challenges for the global model to perform well on all device's local data. When we compare Figure 12a and Figure 12b, we see the striking difference between `FedBC` and FedAvg. For FedAvg, the global model performs poorly on the majority of min users as their test accuracy is 0%. For `FedBC`, there is no performance gap for most points. This shows

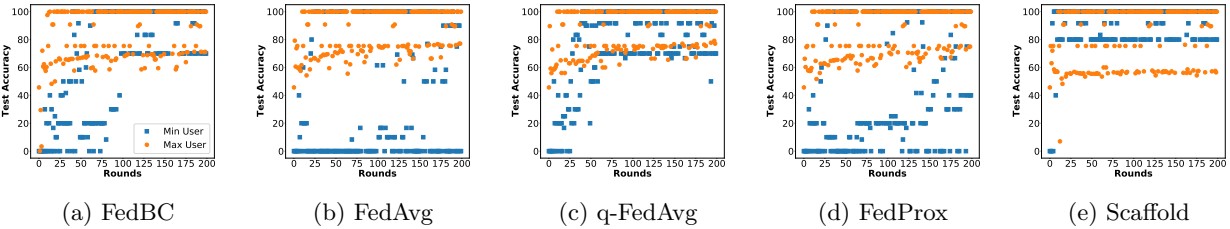

(a) FedBC    (b) FedAvg    (c) q-FedAvg    (d) FedProx    (e) Scaffold

Figure 9: Test accuracy of global model on min and max (defined in Figure 3) device's local data at each round of communication ($E = 1$).

that the global model of `FedBC` is able to perform well on the device's local data despite the presence of high dissimilarity between local and global models.

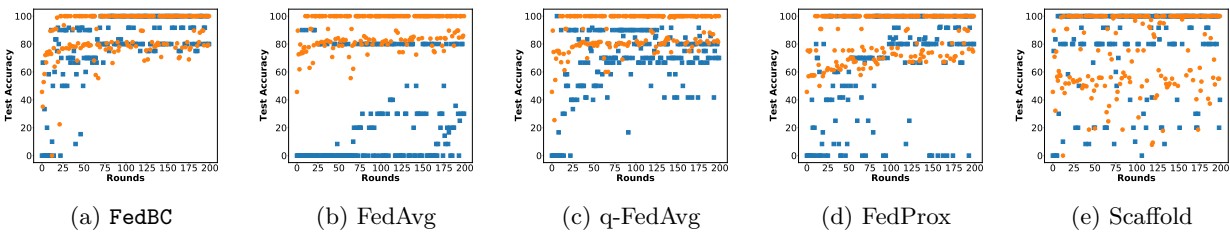

(a) FedBC    (b) FedAvg    (c) q-FedAvg    (d) FedProx    (e) Scaffold

Figure 10: Test accuracy of global model on min and max (defined in Figure 3) device's local data at each round of communication ($E = 10$).

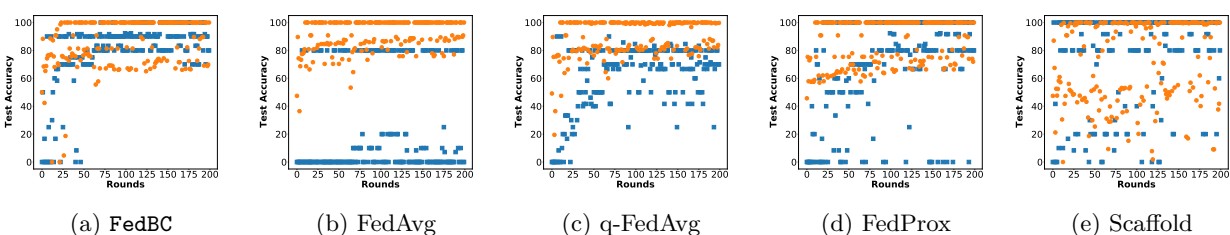

(a) FedBC    (b) FedAvg    (c) q-FedAvg    (d) FedProx    (e) Scaffold

Figure 11: Test accuracy of global model on min and max (defined in Figure 3) device's local data at each round of communication ($E = 25$).

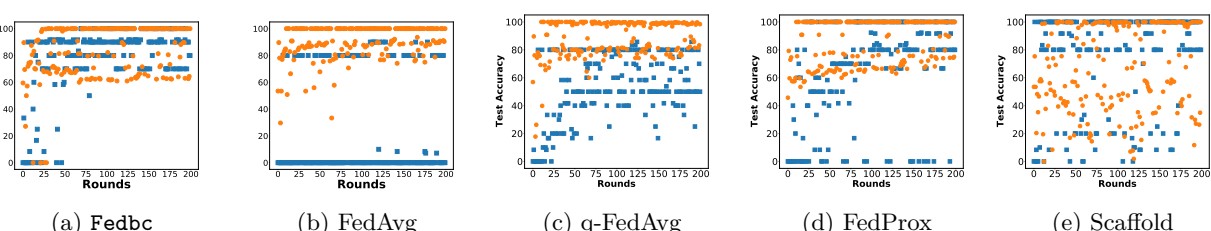

(a) Fedbc    (b) FedAvg    (c) q-FedAvg    (d) FedProx    (e) Scaffold

Figure 12: Test accuracy of global model on min and max (defined in Figure 3) device's local data at each round of communication ($E = 50$).

# F   Additional Experiments on Real Datasets

## F.1   Experiments on CIFAR-10 Dataset

Table 6 shows the CIFAR-10 classification results when $E = 1$. We observe that `FedBC` performs well for $C = 1$ and $C = 3$. For example, for $C = 1$ with 1.3 as a power law exponent, `FedBC` outperforms all other algorithms by more than 3%. Figure 13 shows the coefficient of min and max users/devices for different $C$s. Similar to Figure 8, we *observe the equal performance in treating devices* of different data sizes when Lagrangian multipliers are used to aggregate local models.

Figure 14 and Figure 15 show the variance of test accuracy over the device's local data for different power law exponents and $C$s when $E = 1$ and $E = 5$ respectively. We observe that `FedBC` achieves high uniformity in test accuracy in these heterogeneous environments compared to other algorithms. For example, for $E = 1$ and $C = 3$ in Figure 14(c), `FedBC` has the smallest variance of test accuracy for all power law exponents. This is consistent with what was observed in Figure 3d on the synthetic dataset. In the main body of the paper, Table 2 (for $E = 5$) and Table 3 shows the global and local performance of `FedBC` respectively for all $C$s and power law exponents $1.1, 1.2, 1.3, 1.4$, and $1.5$. In Table 2, we first observe that `FedBC` has the best global performance for all exponents in the most challenging case when $C = 1$. For example, it outperforms FedAvg by 4.59% and 4.41% when $E = 1.2$ and $E = 1.4$ respectively. We also report the global performance for $E = 1$ in Table 6.

On the local performance side in Table 3, we observe that Per-`FedBC` has better local performance than others when power law exponents are $1.2, 1.3, 1.4$, and $1.5$ for $C = 2, 3$. For power law exponent is 1.1, Per-FedAvg has the best local performance. We also observe significant performance improvements of `FedBC`-FineTune over `FedBC`. For example, when the power law exponent is 1.3, `FedBC`-FineTune outperforms `FedBC` by 16.99% in test accuracy for $C = 3$. In general, `FedBC` has the best global accuracy among all algorithms. Its personalization performance can be largely improved when combined with Fine-Tuning or MAML-type training as in Algorithm 3.

| Classes | Algorithm | Power Law Exponent | | | | |
|---------|-----------|------|------|------|------|------|
| | | 1.1 | 1.2 | 1.3 | 1.4 | 1.5 |
| $C = 1$ | FedAvg | $45.13 \pm 1.34$ | $47.40 \pm 1.57$ | $54.56 \pm 1.72$ | $54.60 \pm 2.45$ | $58.88 \pm 1.34$ |
| | q-FedAvg | $\mathbf{45.66 \pm 1.83}$ | $46.75 \pm 2.05$ | $54.27 \pm 2.29$ | $54.49 \pm 2.12$ | $\mathbf{59.25 \pm 1.75}$ |
| | FedProx | $45.05 \pm 1.38$ | $48.37 \pm 2.03$ | $54.94 \pm 1.11$ | $54.82 \pm 0.62$ | $58.88 \pm 1.28$ |
| | Scaffold | $34.94 \pm 1.14$ | $34.57 \pm 1.72$ | $28.82 \pm 8.60$ | $24.18 \pm 6.22$ | $25.21 \pm 6.85$ |
| | FedBC | $45.63 \pm 1.37$ | $\mathbf{49.13 \pm 1.66}$ | $\mathbf{58.21 \pm 1.51}$ | $\mathbf{55.43 \pm 1.60}$ | $58.21 \pm 1.61$ |
| $C = 2$ | FedAvg | $51.82 \pm 1.02$ | $53.86 \pm 2.16$ | $54.07 \pm 4.55$ | $38.45 \pm 3.07$ | $\mathbf{56.55 \pm 3.83}$ |
| | q-FedAvg | $\mathbf{52.14 \pm 0.47}$ | $\mathbf{54.55 \pm 1.46}$ | $53.67 \pm 4.52$ | $\mathbf{54.45 \pm 6.46}$ | $56.41 \pm 2.53$ |
| | FedProx | $51.78 \pm 0.78$ | $53.48 \pm 1.56$ | $53.93 \pm 3.15$ | $37.01 \pm 1.77$ | $58.22 \pm 3.09$ |
| | Scaffold | $47.98 \pm 0.51$ | $43.98 \pm 1.50$ | $32.38 \pm 2.29$ | $37.70 \pm 7.28$ | $23.54 \pm 8.44$ |
| | FedBC | $51.83 \pm 1.32$ | $52.59 \pm 1.60$ | $\mathbf{56.11 \pm 1.52}$ | $50.41 \pm 5.80$ | $\mathbf{56.55 \pm 2.81}$ |
| $C = 3$ | FedAvg | $57.31 \pm 1.10$ | $49.42 \pm 4.00$ | $53.19 \pm 1.36$ | $55.53 \pm 1.39$ | $54.84 \pm 1.37$ |
| | q-FedAvg | $59.22 \pm 0.70$ | $55.26 \pm 3.87$ | $55.39 \pm 3.20$ | $56.88 \pm 2.12$ | $58.51 \pm 3.68$ |
| | FedProx | $58.17 \pm 1.55$ | $49.37 \pm 3.59$ | $52.78 \pm 1.82$ | $56.36 \pm 1.19$ | $56.57 \pm 0.83$ |
| | Scaffold | $55.18 \pm 1.47$ | $49.86 \pm 1.96$ | $33.71 \pm 2.14$ | $34.74 \pm 2.68$ | $35.53 \pm 3.78$ |
| | FedBC | $58.32 \pm 2.23$ | $\mathbf{56.37 \pm 2.90}$ | $\mathbf{58.97 \pm 1.32}$ | $\mathbf{61.09 \pm 0.95}$ | $\mathbf{61.17 \pm 0.86}$ |

Table 6: CIFAR-10 top-1 classification accuracy ($E = 1$).

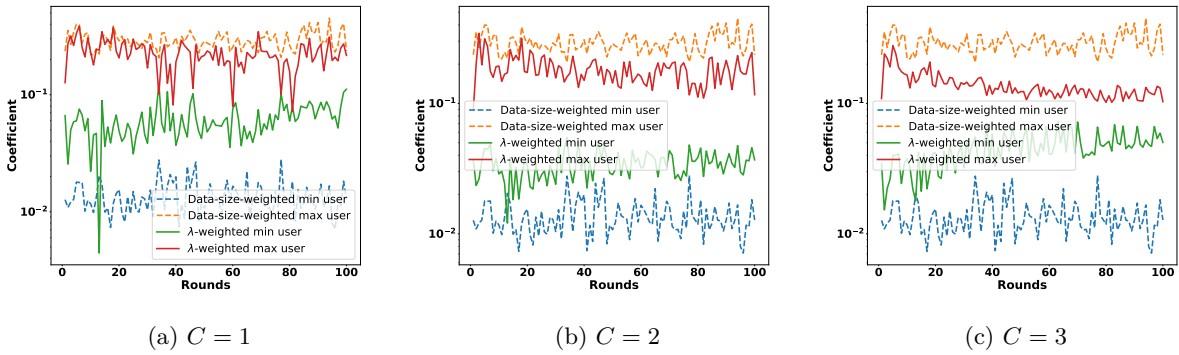

Figure 13: Coefficient of min and max device based on either Lagrangian multiplier or data size when computing the global model for $C = 1$ (a), 2 (b), and 3 (c). We set power law exponent to be 1.2 and $E = 5$.

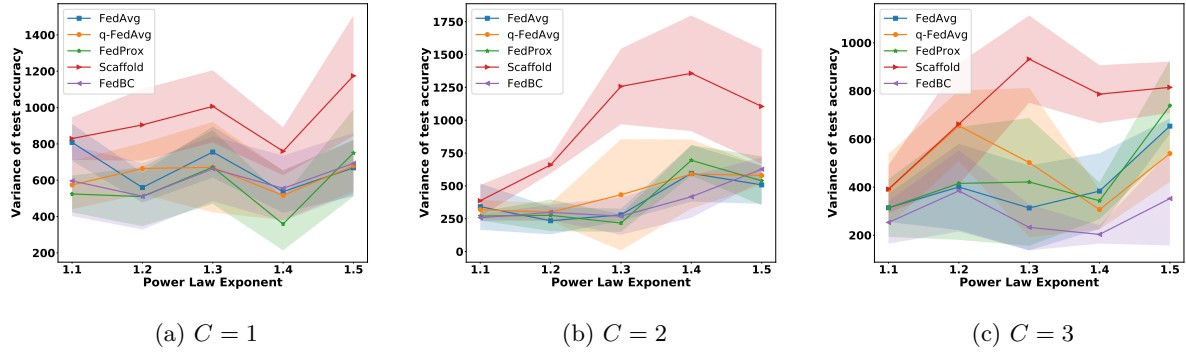

Figure 14: Variance of test accuracy of global model over device's local data against different power law exponents for $C = 1, C = 2$, and $C = 3$ (shaded area shows standard deviation, $E = 1$).

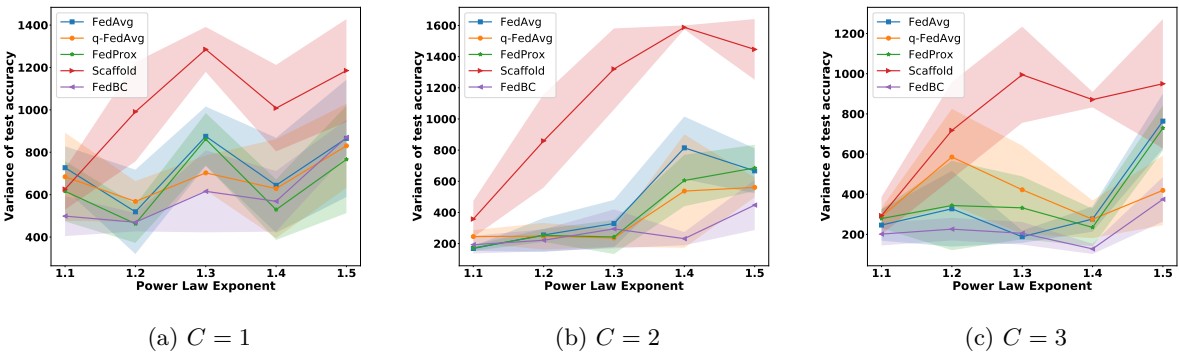

Figure 15: Variance of test accuracy of global model over device's local data against different power law exponents for $C = 1, C = 2$, and $C = 3$ (shaded area shows standard deviation, $E = 5$).

## F.2 Experiments on MNIST Dataset

We present the global performance results for the MNIST classification task in Table 7 and Table 8 for $E = 1$ and $E = 5$, respectively. The results are consistent with those of CIFAR-10. We again observe that FedBC has good performance for different $C$s and power law exponents. For example, when $C = 1$

and 1.3 as power law exponent, `FedBC` outperforms FedAvg by 3% for both $E = 1$ and $E = 5$. We also observe that `FedBC` outperforms others when exponents are 1.2, 1.4, and 1.5 for $C = 1, 3$ with $E = 5$. These results demonstrate that `FedBC` achieves the best global performance for different datasets under challenging heterogeneous environments.

| Classes | Algorithm | Power Law Exponent | | | | |
|---|---|---|---|---|---|---|
| | | 1.1 | 1.2 | 1.3 | 1.4 | 1.5 |
| $C = 1$ | FedAvg | $93.71 \pm 0.25$ | $93.05 \pm 0.26$ | $90.75 \pm 0.64$ | $94.38 \pm 0.18$ | $94.36 \pm 0.37$ |
| | q-FedAvg | $\mathbf{94.15 \pm 0.11}$ | $93.25 \pm 0.40$ | $92.36 \pm 0.95$ | $94.59 \pm 0.15$ | $94.23 \pm 0.50$ |
| | FedProx | $93.88 \pm 0.31$ | $93.79 \pm 0.45$ | $91.69 \pm 0.51$ | $94.39 \pm 0.15$ | $94.39 \pm 0.39$ |
| | Scaffold | $94.14 \pm 0.14$ | $92.79 \pm 0.11$ | $93.62 \pm 0.25$ | $94.65 \pm 0.35$ | $94.49 \pm 0.21$ |
| | FedBC | $94.11 \pm 0.28$ | $\mathbf{94.75 \pm 0.37}$ | $\mathbf{93.86 \pm 0.41}$ | $\mathbf{94.79 \pm 0.41}$ | $\mathbf{95.04 \pm 0.30}$ |
| $C = 2$ | FedAvg | $95.97 \pm 0.15$ | $95.33 \pm 0.39$ | $95.39 \pm 0.27$ | $95.50 \pm 0.29$ | $95.39 \pm 0.34$ |
| | q-FedAvg | $95.87 \pm 0.14$ | $95.63 \pm 0.27$ | $\mathbf{95.71 \pm 0.30}$ | $95.53 \pm 0.32$ | $\mathbf{95.81 \pm 0.46}$ |
| | FedProx | $95.85 \pm 0.18$ | $95.39 \pm 0.33$ | $95.38 \pm 0.34$ | $95.46 \pm 0.28$ | $95.38 \pm 0.37$ |
| | Scaffold | $95.93 \pm 0.16$ | $\mathbf{96.31 \pm 0.09}$ | $94.70 \pm 0.18$ | $94.73 \pm 0.11$ | $94.59 \pm 0.18$ |
| | FedBC | $\mathbf{96.06 \pm 0.16}$ | $95.03 \pm 0.31$ | $95.62 \pm 0.28$ | $\mathbf{95.69 \pm 0.27}$ | $95.79 \pm 0.28$ |
| $C = 3$ | FedAvg | $96.69 \pm 0.24$ | $96.12 \pm 0.42$ | $95.78 \pm 0.14$ | $96.51 \pm 0.06$ | $96.23 \pm 0.19$ |
| | q-FedAvg | $96.65 \pm 0.10$ | $96.54 \pm 0.22$ | $96.11 \pm 0.26$ | $96.56 \pm 0.19$ | $\mathbf{96.43 \pm 0.18}$ |
| | FedProx | $96.75 \pm 0.12$ | $95.86 \pm 0.38$ | $95.77 \pm 0.17$ | $96.53 \pm 0.07$ | $96.20 \pm 0.17$ |
| | Scaffold | $\mathbf{97.04 \pm 0.09}$ | $\mathbf{96.99 \pm 0.21}$ | $94.40 \pm 0.11$ | $94.58 \pm 0.07$ | $94.09 \pm 0.17$ |
| | FedBC | $96.83 \pm 0.13$ | $96.88 \pm 0.13$ | $\mathbf{96.46 \pm 0.26}$ | $\mathbf{96.72 \pm 0.16}$ | $96.40 \pm 0.13$ |

Table 7: MNIST top-1 classification accuracy ($E = 1$).

| Classes | Algorithm | Power Law Exponent | | | | |
|---|---|---|---|---|---|---|
| | | 1.1 | 1.2 | 1.3 | 1.4 | 1.5 |
| $C = 1$ | FedAvg | $93.36 \pm 0.33$ | $93.26 \pm 0.30$ | $90.52 \pm 0.96$ | $94.37 \pm 0.31$ | $94.34 \pm 0.41$ |
| | q-FedAvg | $93.13 \pm 0.42$ | $93.20 \pm 0.29$ | $92.10 \pm 1.56$ | $94.64 \pm 0.19$ | $94.30 \pm 0.49$ |
| | FedProx | $93.24 \pm 0.55$ | $92.09 \pm 0.76$ | $90.52 \pm 0.97$ | $94.39 \pm 0.36$ | $94.45 \pm 0.57$ |
| | Scaffold | $\mathbf{94.92 \pm 0.12}$ | $93.88 \pm 0.34$ | $90.43 \pm 2.66$ | $77.00 \pm 3.95$ | $76.19 \pm 4.34$ |
| | FedBC | $93.82 \pm 0.41$ | $\mathbf{94.01 \pm 0.33}$ | $\mathbf{93.18 \pm 0.54}$ | $\mathbf{94.98 \pm 0.30}$ | $\mathbf{95.33 \pm 0.27}$ |
| $C = 2$ | FedAvg | $95.62 \pm 0.19$ | $95.59 \pm 0.37$ | $95.66 \pm 0.27$ | $95.73 \pm 0.42$ | $95.62 \pm 0.45$ |
| | q-FedAvg | $95.65 \pm 0.11$ | $95.78 \pm 0.20$ | $\mathbf{96.05 \pm 0.26}$ | $95.86 \pm 0.41$ | $\mathbf{96.20 \pm 0.36}$ |
| | FedProx | $95.97 \pm 0.19$ | $95.60 \pm 0.39$ | $95.76 \pm 0.29$ | $95.74 \pm 0.47$ | $95.56 \pm 0.55$ |
| | Scaffold | $\mathbf{96.68 \pm 0.05}$ | $\mathbf{96.40 \pm 0.13}$ | $93.93 \pm 0.99$ | $91.94 \pm 1.66$ | $76.46 \pm 5.20$ |
| | FedBC | $96.15 \pm 0.17$ | $95.80 \pm 0.29$ | $95.91 \pm 0.42$ | $\mathbf{96.03 \pm 0.30}$ | $96.02 \pm 0.32$ |
| $C = 3$ | FedAvg | $97.29 \pm 0.23$ | $96.68 \pm 0.32$ | $96.21 \pm 0.17$ | $96.80 \pm 0.12$ | $96.50 \pm 0.15$ |
| | q-FedAvg | $97.43 \pm 0.05$ | $97.02 \pm 0.22$ | $96.69 \pm 0.26$ | $96.92 \pm 0.10$ | $96.58 \pm 0.23$ |
| | FedProx | $97.14 \pm 0.16$ | $96.69 \pm 0.30$ | $96.21 \pm 0.12$ | $96.86 \pm 0.08$ | $96.50 \pm 0.11$ |
| | Scaffold | $\mathbf{97.75 \pm 0.06}$ | $97.08 \pm 0.22$ | $95.84 \pm 0.27$ | $93.51 \pm 0.51$ | $89.44 \pm 1.96$ |
| | FedBC | $97.23 \pm 0.13$ | $\mathbf{97.11 \pm 0.14}$ | $\mathbf{96.80 \pm 0.12}$ | $\mathbf{97.10 \pm 0.19}$ | $\mathbf{96.87 \pm 0.24}$ |

Table 8: MNIST top-1 classification accuracy ($E = 5$).

## F.3 Experiments on Shakespeare Dataset

For the Shakespeare dataset, we report the global model performance for the next-character prediction task in Table 9. The top row presents the mean prediction accuracy. We observe algorithms have very similar performance when $E = 1$ or 5 except for Scaffold. For $E = 1$, `FedBC` outperforms q-FedAvg and Scaffold, but performs slightly worse than FedAvg and FedProx. For $E = 5$, `FedBC` has the best performance. The second row presents the variance of test accuracy over device's local dataset. We observe Scaffold has the smallest variance. The variance of `FedBC` is larger than Scaffold but smaller than others when $E = 1$; it is smaller

than FedAvg but greater than others when $E = 5$. In general, the performance of `FedBC` on the Shakespeare dataset is competitive when compared against others.

| Epoch | Algorithms | | | | |
|-------|-----------|-----------|-----------|-----------|-----------|
| | FedAvg | q-Fedavg | FedProx | Scaffold | FedBC |
| 1 | $52.10 \pm 0.20$ | $51.95 \pm 0.30$ | $52.07 \pm 0.16$ | $32.13 \pm 4.13$ | $51.97 \pm 0.34$ |
| | $79.75 \pm 8.91$ | $84.43 \pm 15.20$ | $85.19 \pm 6.95$ | $60.93 \pm 4.78$ | $79.45 \pm 7.02$ |
| 5 | $49.21 \pm 0.20$ | $50.44 \pm 0.20$ | $50.58 \pm 0.25$ | $32.50 \pm 3.64$ | $50.69 \pm 0.29$ |
| | $91.10 \pm 4.22$ | $81.05 \pm 10.46$ | $81.60 \pm 10.13$ | $56.54 \pm 8.65$ | $85.44 \pm 13.01$ |

Table 9: Next character prediction is based Shakespeare dataset. The first row shows the prediction accuracy of the test dataset. The second row shows the variance of test accuracy over the device's local dataset. The $\pm$ indicates the standard deviation. The total number of devices is 138.

