# OpenReview forum: "\texttt{FedBC}: Federated Learning Beyond Consensus"
_TMLR — Rejected by TMLR_

### Review · Reviewer_j379 · 2023-07-03

**Summary Of Contributions:**

The paper makes the following contributions:

- An approach to federated learning with a relaxed consensus constraint where each local model $x_i$ is required to be close to the global model $z$ as $\Vert x_i - z\Vert_2^2 \le \gamma_i$ for some parameter $\gamma_i$. This framework provides a global model $z$ and personalized models $x_i$ per client.
- A novel Lagrangian optimization algorithm to optimize this objective. The resulting algorithm is well-suited to federated learning in the sense that (a) the server broadcasts the global model on a  sampled set of clients in each round, (b) each client computes local model updates with GD/SGD and sends them to the server along with updated dual variables, and (c) the client updates are aggregated with a weighted average.
- A convergence analysis of the proposed algorithm in the non-convex case. The proposed algorithm is competitive (although several algorithms perform within one standard deviation).
- Extensive numerical experiments.

**Audience:**

Yes

**Claims And Evidence:**

No

**Requested Changes:**

It is essential that the authors fix the potential technical issues with the convergence analysis (and improve the clarity). I am not convinced of the correctness of the proofs as they are.

In addition, I consider the following changes as required for acceptance of the paper for publication (see above for details):
- Fully describe the algorithm
- Improve the clarity of presentation of the convergence results
- Give more details about the selection of $\gamma_i$

I strongly encourage the authors to address the other concerns described above as well since that would greatly strengthen the paper.

**Strengths And Weaknesses:**

# Strengths

- The paper presents a novel algorithm for solving a constrained optimization problem with a Lagrangian approach to give an algorithm well-suited to the federated setting. I view this as a major strength of this paper.
- The approach yields both a global model and personalized models per client.
- The numerical experiments are comprehensive with detailed comparisons to state-of-the-art baselines. The paper also gives some interesting analyses of the properties of the proposed algorithm such as the evolution of the weights over training and the behavior w.r.t. amount of data per client.
- The paper is very well-written and easy to follow (except the convergence results, see below).

# Weaknesses

**The algorithm is not fully specified**:  In Algorithm 1, what happens on clients that do not participate? I assume that $x_i^{t+1} = x_i^t$ and $\lambda_i^{t+1} = \lambda_i^t$ in my further comments. Note that there is a less practical alternative where the local updates are performed on the device but they are not aggregated; this ambiguity may have been the root cause of the potential technical issues I describe next.

**Potential technical errors in the proofs**: I think there is some technical issue with the expectations. If I am mistaken, please point if out.
- In Eq. (50): $x_j^{t+1}$ should ideally depend on $\mathcal{S}_t$ (if $j \notin \mathcal{S}_t$, then $x_j^{t+1} = x_j^t$, else it is updated). I belive Eq. (50) is incorrect for this reason.
- The authors might find the technique of virtual full participation to be helpful as a similar problem is tackled there: see Appendix A.2 of [Pillutla et al. (ICML '22)](https://proceedings.mlr.press/v162/pillutla22a/pillutla22a.pdf).
- A similar issue also shows up in the proof of Lemma B.1. Eq. (41) is correct only for $i \in \mathcal{S}_t$ but incorrect for $i \notin \mathcal{S}_t$. Eq. (42) must therefore take an expectation over all uniform samples of size $S$ from $N$. This changes needs to be propagated throughout.
- On a related note, what is the underlying cause of the randomness for each expectation? It is not clear at all and makes the paper harder to read and is potentially a reason behind the technical issues above. The rigorous way is to define a filtration and specify the conditional expectations clearly.

**Clarity of presenting convergence results**: Several details are missing in the presentation of the results.
- The constraint set $\Lambda$ of the dual variables is not specified explicitly. These details appear in the middle of the proofs in the appendix but they should be explained clearly in the description of the algorithms and the convergence results.
- The _problem-dependent constants_ in Theorem 4.5, Corollary 4.6, Lemma B.2 are hidden in the big-O notation. It is ok to hide these constants when discussing the results in text but it would be nice to see them be explicitly specified in theorem statements. (It is ok to omit absolute constants and logarithmic factors).
- Similarly, the constant $M_1$ (Lemma B.1) should be defined upfront.
- The proofs in Appendix B use $L$ (rather than $L_i$), see e.g. Eq. 35. I could not find the definition for it.
- What is the total cost of computation for the local updates per client? Also, what is total number of gradient computations required for a certain error? How does that compare to FedAvg?

**Selection of $\gamma_i$'s**: The paper proposes a heuristic to automatically adapt the value of the hyperparameter of $\gamma_i$. This is very interesting but it is not developed adequately. What is the original optimization problem that leads to this heuristic? Is that optimization problem well-defined? What if $\gamma_i = \infty$ the optimal solution? What is the learning rate used for the $\gamma_i$-update in practice? How is it tuned?

**Homogoenity of constants**: There is a term of the form $(1 + L_i)$ that appears in Appendix B. The constant $L_i$ has "units" associated with it and it does not make sense to add 1 to it. For instance, if I change the description of the data from kilometers to miles for example, the new problem is exactly equivalent to the older one up to scaling ($L_i$ gets scaled by a constant). The solution is simple: change the condition $\lambda_i^t \ge 1 + L_i$ to $\lambda_i \ge 2 L_i$.

**Misuse of the term "fairness"**: The term fairness in machine learning has very rigorous definitions such as equalized odds, where the goal is that sensitive information does not “unfairly” influence the outcomes of learning methods (see e.g. [Barocas et al.](https://fairmlbook.org/)). The notion of "fairness" used in this work (coming from Li et al.) does not match this more established definitions and can be confusing to the broader community. I would therefore recommend the authors to use the term "performance disparity" or "per-client variations" or similar terms. On a related note, it would be much more interpretable to present the results of Figure 3(d) in terms of the standard deviation rather than the variance.

**Minor typos or comments**: I suggest the authors to thoroughly proofread all calculations and text. Here is a non-exhaustive list of typos to fix:
- Eq. 39: should the $\alpha G^2$ term be $\alpha G$ instead?
- Eq. 40 and 41 are repeated.
- Eq. 58: should the $\sqrt{\epsilon}$ term be $2 / (1 + L_i)$?
- The text before Eq. 57: Cauchy-Schwarz -> triangle inequality
- Eq. 63: A factor of $S(N-1)/N$ seems to be missing in the first inequality.

---

### Review · Reviewer_TbLe · 2023-07-03

**Summary Of Contributions:**

This paper proposes to relax the consensus requirement in traditional federated learning algorithms. The paper firstly reformulates the loss function used in FL and relaxes it with a soft consensus constraint. After that, the paper uses Lagrangian relaxation to transform the constrained optimization problem into the form of primal-dual optimization, which naturally gives rise to a novel principled FL algorithm.

**Audience:**

Yes

**Broader Impact Concerns:**

There is no ethical implication that I can see.

**Claims And Evidence:**

Yes

**Requested Changes:**

Please refer to the three points I listed under "Weaknesses" above. I would appreciate some responses discussing these three points, and I believe adding such relevant discussions to the paper will make the paper more complete and stronger.

**Strengths And Weaknesses:**

Strengths:
- The paper has taken a natural and principled approach to derive the final algorithm. Specifically, the paper firstly reformulates standard FL (equation 3), and then relaxes the hard consensus constraints into soft constraints (equation 7), after which standard Lagrangian relaxation is applied to derive natural updating rules which fit nicely with global and local optimization respectively.
- Last paragraph of Section 3: this paragraph gives a nice discussion regarding how the proposed algorithm, despite being developed through a novel route via Lagrangian relaxation, is related various existing works. It also serves to justify that the proposed algorithm is reasonable and principled.
- Overall, the experimental results are promising, since the proposed method can indeed achieve better global and local performances (after the addition of some heuristics, which look reasonable to me).
- The heuristic used to select $\gamma_i$ is interesting to me, and it leads to some interesting insights on the behavior of the proposed algorithm

Weaknesses:
- There seems to be a discrepancy between the theoretical and empirical results regarding the dependency of the global performance on the $\gamma_i$'s. Firstly, Figure 2 (as well as the main experiments which all use nonzero $\gamma_i's$) show that when $\gamma_i$ is small, the global performance is improved with a larger value of $\gamma_i$. However, the theoretical results as shown in Theorem 4.5 (the last term) suggests that the global convergence becomes better as the $\gamma_i$'s become smaller. What's the reason behind this discrepancy?
- Regarding the fairness achieved by the proposed algorithm. The experimental results indeed show that the proposed algorithm can achieve better fairness, however, I think there should be an in-depth discussion as to why the proposed algorithm can do this. This is because during the design process of the proposed algorithm (Sections 2 and 3), I don't see any particular design choice specifically targeted at achieving a fairer solution. Also, the fairness capability of the proposed algorithm is also not reflected in the convergence analysis in Section 4.
- Theoretical results in Section 4: Overall I like the theoretical section, but I feel that some of the dependencies in the theoretical results should be further discussed. One example is the dependency of the global convergence on $\gamma_i$ I mentioned above. Also, Theorem 4.5 has a dependency of $\mathcal{O}(\alpha)$ which suggests that to achieve a faster convergence, $\alpha$ should be $0$ (which clearly isn't desirable); how should we interpret this? In addition, how does device sampling affect the convergence results? I don't see the dependency of the convergence on the set of sampled devices $\mathcal{S}_t$.


Minor points:
- Figure 1, the legends for the two arrows at the bottom right corner: are these two legends reversed?
- Abstract, line 5: an improve -> improves
- Theorem 4.5 and Corollary 4.6: I guess it should be "Under assumptions 4.1-4.4"?

---

### Review · Reviewer_dGvW · 2023-08-22

**Summary Of Contributions:**

The paper proposes a new method for personalised FL wherein they modify the global objective being minimized in a way that each client can have its own personal model which is “close” to the global model. The proposed objective is solved using a primal-dual method. And a convergence analysis of the method is provided.

**Audience:**

Yes

**Broader Impact Concerns:**

In my opinion, there are no broader impact concerns from this work.

**Claims And Evidence:**

Yes

**Requested Changes:**

**Questions**:

1. If the clients are training local models, does not it make more sense to obtain the results like in Figure 3 and 4 for the local client models as opposed to the global model?

2. How realistic is the Assumption 4.3 in real heterogeneous settings?

3. There are other methods in literature like [1], which perform divergence aware updates on the local models, is it possible to compare solving the proposed optimization problem against performing divergence aware updates?

[1]Divergence-aware Federated Self-Supervised Learning

**Strengths And Weaknesses:**

**Strengths**:

1. The new formulated objective is interesting as it is a general version of the existing methods.

2. A convergence guarantee of the method is provided.

**Weaknesses**:

1. Literature review is limited.

2. Some details about the experiments like how the data is partitioned, the degree of heterogeneity etc. are missing.

---

### Review · Reviewer_TTpv · 2023-08-23

**Summary Of Contributions:**

This paper studies the federated learning problem, where heterogeneous agents want to improve both local and global performance. The authors propose a novel algorithm, named Federated Learning Beyond Consensus (FedBC), that takes a tolerance parameter $\gamma_i$ (for client $i$) as input and controls the proximity between local and global models. They have also shown that FedBC converges to the first-order stationary points with state-of-the-art rates. To evaluate the performance of the proposed FedBC algorithm, the authors considered different settings for experiments on different datasets like synthetic, MNIST, CIFAR-10, and Shakespeare.

**Audience:**

Yes

**Broader Impact Concerns:**

I do not find any ethical concerns.

**Claims And Evidence:**

Yes

**Requested Changes:**

1. Notation clarity after Eq. (1): Clearly define what variables 'x' and $\zeta_i$ represent.

2. Explain why heterogeneous agents still want to improve global performance instead of focusing only on local performance.

 3. Clearly explain why optimizing the weights of local models in the global objective may not be a good idea compared to using the objective defined in Eq. (7). For example, in Figure 1, if one optimizes weights for $x_1^*$ and $x_2^*$, one may get the same $z^*$ as after solving Eq. (7).

**Strengths And Weaknesses:**

#### **Strengths of paper:**
1. This paper considers a practical federated learning problem with heterogeneous agents. In this problem, agents not only want to improve the global model but also wish to improve the local model.

2. To improve both local and global performance, the authors proposed an algorithm FeDBC that uses a tolerance parameter to control proximity (which can viewed as giving different weightage to local and global models) between local and global models. FedBC uses the primal-duel method to solve the objective function (defined in Eq. (7)).

3. The authors have considered different practical settings and datasets for the detailed performance evaluation of their proposed algorithm.

#### **Weaknesses of paper:**
The only weakness of the proposed algorithm is it needs tolerance parameter $\gamma_i$ for each client as input. It is unclear if there is any principled way to select these parameters for any given FL problem.

---

### Decision · Action_Editors · 2023-10-10

**Recommendation:** Reject

**Comment:**

In this paper, the authors investigate the problem of federated learning, and propose to relax the consensus constraint to improve both local and global performance. Specifically, they replace the consensus constraint with the proximity between the local and the global model controlled by a tolerance parameter. Then, they develop a novel Federated Learning Beyond Consensus (FedBC) algorithm, and present empirical results to demonstrate its advantage.

The proposed algorithm shows promise for federated learning with heterogeneous agents. Nevertheless, a reviewer has highlighted a few technical concerns within the analysis. Consequently, the manuscript, in its current form, is not suitable for publication. I recommend that the authors address these concerns and consider submitting a revised version.

**Audience:**

Yes

**Claims And Evidence:**

Reviewer j379 highlighted technical issues within the proof. While the author has addressed some of these concerns in the response, there are outstanding issues as listed by Reviewer j379.

**Resubmission Of Major Revision:**

The authors may consider submitting a major revision at a later time.